# E$^2$-RAG: Towards Editable Efficient RAG by Editing Compressed KV Caches

**Tongxu Luo**[1][*]  **Wenyu Du**[2][*]   **Hanwen Hao**[3]   **Min Zhang**[4]   **Hao Yang**[4]
**Benyou Wang**[1][†]
[1]The Chinese University of Hong Kong, Shenzhen
[2]The University of Hong Kong   [3]Beihang University   [4]Huawei
tongxuluo@gmail.com   wenyu.du@dualityrl.com   wangbenyou@cuhk.edu.cn

## Abstract

Retrieval-Augmented Generation (RAG) demonstrates remarkable capabilities for enhancing the performance of Large Language Models (LLMs) by integrating external knowledge. RAG introduces additional computations due to the extra retrieved context. To improve efficiency, recent studies propose compressing chunk tokens into compact forms, such as key-value (KV) caches. However, maintaining these compressed KV caches in an updated state presents a significant challenge, undermining the primary goal of RAG: acquiring up-to-date knowledge. In this work, we propose **E$^2$-RAG**, the first **E**ditable **E**fficient-**RAG** method designed to efficiently edit compressed KV caches for knowledge updates in fast updating scenarios. E$^2$-RAG features an encoder-decoder architecture as efficient RAG module, along with an additional editor. The encoder-decoder compresses chunk tokens into KV caches and generates responses. The editor takes old KV caches and new knowledge tokens as inputs, enabling efficient updates to the KV caches. To formalize knowledge updating, we define three operations: **INSERT**, **DELETE**, and **UPDATE**. We create three sets of datasets for each operation. Through extensive experiments, E$^2$-RAG achieves nearly **40x faster** editing compared to recomputing KV caches while maintaining **3x faster** generation efficiency than standard RAG, with a performance downgrade of 1%-5%. We also conduct ablation studies such as multi-turn editing, multi-chunk capability, and knowledge conflicts, to explore the capabilities of E$^2$-RAG. Our code, datasets, and models are available at https://github.com/tongxuluo/e2rag.

## 1 Introduction

Large Language Models (LLMs) exhibit impressive text generation capabilities; however, their knowledge is limited to the dataset used during pre-training. Retrieval-Augmented Generation (RAG) emerges as a promising method to address this limitation by enabling the retrieval of new knowledge from an external database (Borgeaud et al., 2022; Lewis et al., 2020; Kandpal et al., 2023). This knowledge database typically consists of numerous short chunks. When a user inputs a query, relevant chunks are fetched and combined with the user's query to generate the answer (Lewis et al., 2020; Tao et al., 2021; Mao et al., 2024).

A critical challenge in standard RAG systems lies in the substantial computational overhead introduced by processing additional retrieved chunks in input prompts (Tay et al., 2020; Yu et al., 2024). Recent efforts in efficient RAG focus on compressing text chunks into model-specific intermediate representations, such as key-value (KV) caches (Lu et al., 2024; Sun et al., 2024). This compression enables LLMs to process knowledge through these compact representations rather than raw text, significantly accelerating generation speeds.

---

[*] Equal Contributions.
[†] Corresponding Author.

However, world knowledge is constantly evolving. For scenarios such as news and finance, which strongly rely on the timeliness of knowledge, maintaining up-to-date databases for RAG systems is crucial. However, for standard RAG systems, updating their databases with new world knowledge requires either manual curation or LLM-assisted modification. Efficient RAG systems, in addition, must regenerate compressed representations from edited texts. This process incurs substantial computational costs, as evidenced by Wikipedia's 0.6 million daily edits (Wikimedia Foundation, 2025), which would require approximately 32.8 A100 GPU hours for LLM-assisted modification and KV cache regeneration (see Appendix B).

In this work, we introduce **$E^2$-RAG**, an editable efficient RAG framework that directly modifies compressed knowledge representations. Inspired by fundamental database operations (ISO/IEC, 2023), we define three core edit operations: **INSERT**, **DELETE**, and **UPDATE**. The architecture incorporates an *efficient RAG module* and an *editor module* for modifying KV caches. The editor is a frozen LLM equipped with trainable LoRA (Hu et al., 2021) and an editing embedding, which generates offset KV caches to update the old KV cache based on the three operations.

We evaluate $E^2$-RAG using modified versions of QA datasets (HotpotQA (Yang et al., 2018), ASQA (Stelmakh et al., 2022), SciQ (Welbl et al., 2017), SQuAD (Rajpurkar, 2016), Drop (Dua et al., 2019)) containing 10,000 knowledge edits per operation type. Figure 1 shows one example. We compare $E^2$-RAG against two baselines: a standard RAG (Mao et al., 2024) and our efficient RAG module. The results indicate that $E^2$-RAG is **three times** faster in generation than the standard RAG and achieves an editing speed that is **40 times** faster than the efficient RAG that re-computes KV caches, with only a 1%-5% performance degradation across five QA benchmarks. Furthermore, we extend $E^2$-RAG to multi-turn editing and multi-chunk inference settings and discuss knowledge conflicts.

Our analysis extends to multi-edit scenarios and multi-chunk inference, revealing insights into knowledge conflict resolution during **DELETE** and **UPDATE** operations. As RAG systems increasingly adopt compressed knowledge representations, $E^2$-RAG provides a scalable solution for efficient knowledge maintenance.

Our contributions can be summarized as follows:

1. We focus on the efficient editing of RAG databases in compressed KV form and provide insights into this challenge.

2. We propose $E^2$-RAG, the first editable efficient RAG architecture that achieves both efficiency in editing and inference.

3. We construct three datasets designed to evaluate three core operations: **INSERT**, **DELETE**, and **UPDATE**.

4. We conduct comprehensive experiments evaluating the editability and efficiency of the proposed methods, along with further ablations and discussions.

## 2   RAG and Knowledge Updating

**RAG.**   RAG is a "Retrieve-Read" framework (Gao et al., 2023). The standard RAG system first splits a document into $n$ chunks $\{D_i\}_{i=1}^n$, which are then encoded into embedding vectors $\{E_i\}_{i=1}^n$ using an embedding model. These vectors and chunks construct a database $\mathbb{D} = \{(E_i, D_i)\}_{i=1}^n$. When a user inputs a query $q$, the RAG system converts it into a vector $E_q$ using the same embedding model and computes its similarity with $\{E_i\}$. The top $k$ chunks with the highest similarity, $\{\hat{D}_1, \cdots, \hat{D}_k\}$, are retrieved (Karpukhin et al., 2020). These $k$ chunks, along with $q$, are then input into the LLM to generate the response.

However, a significant drawback of standard RAG is the increased computational costs during the generation phase due to the additional RAG chunks integrated into the input prompts (Khandelwal et al., 2019; Izacard & Grave, 2020; Qin et al., 2023). Consequently, various efficient RAG approaches have been proposed to alleviate this overhead (Yan

et al., 2024). Among these, state-of-the-art methods compress chunk tokens into compact intermediate components in LLMs, such as KV caches (Li et al., 2024b), resulting in databases of the form $\mathbb{D} = \left\{(E_i, (K_i, V_i))\right\}_{i=1}^{n}$. This allows LLMs to bypass processing chunks in their raw text forms, significantly increasing generation speed compared to standard RAG. However, this also means that KV caches cannot be easily edited, which contradicts the goals of RAG. Recomputing these caches could lead to additional computational overhead.

**Knowledge Update.** Knowledge evolves continuously, and an active knowledge base system should track these changes. For example, Wikipedia (Wikimedia Foundation, 2025) maintains a comprehensive editing log that records all editing history. Based on these log entries, we categorize them into three types: adding new knowledge, deleting obsolete knowledge, and replacing existing knowledge [1]. To align with basic SQL (ISO/IEC, 2023) operations, we refer to these three knowledge operations as **INSERT**, **DELETE**, and **UPDATE**. For instance, consider a document about the Olympic Games. During a major Olympic event, the list of medalists will **INSERT** new entries, while some outdated items may be **DELETE**, and certain records will be **UPDATE**.

**Knowledge Update for RAG.** Knowledge updates are straightforward for standard RAG since it can integrate up-to-date texts internally. However, this process is not simple for efficient RAG approaches that utilize compressed KV chunks. The naive method involves using the updated text to regenerate these KV caches for future RAG generation. The standard RAG solution allows for easy editing but is insufficient for generation, while the opposite is true for the latter approach. Thus, instead of the above two solutions, we investigate *whether we can directly edit the existing KV caches for efficient RAG.*

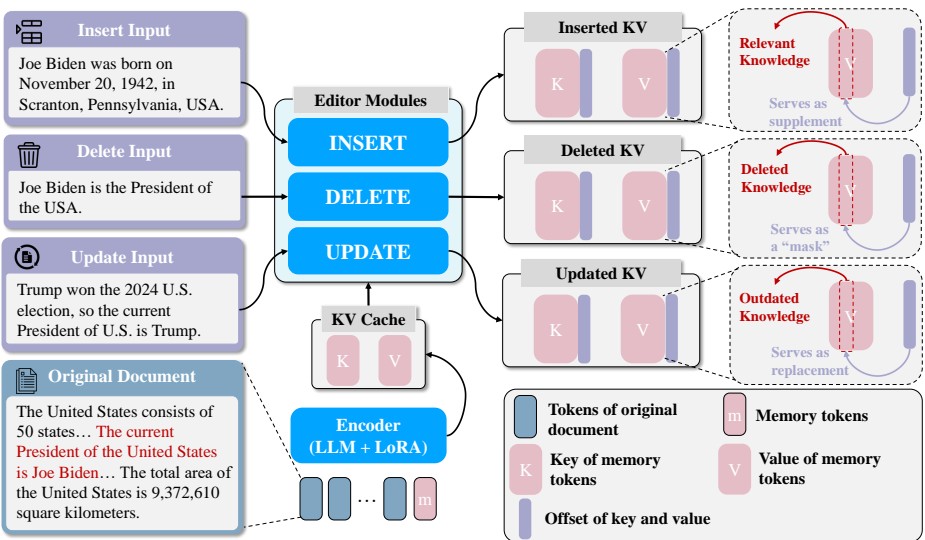

Figure 1: The editor module. We train three separate editors for **INSERT**, **DELETE**, **UPDATE** respectively. And the offset KV caches appends to old KV caches but functions differently.

# 3 E²-RAG

In this section, we detail our method, E²-RAG, which includes its efficient RAG module for pretraining to learn compression, fine-tuning for multi-chunk question answering, and the training of the E²-RAG editor module.

---

[1] We provide the statistics, and analysis of "Machine Learning" category in Appendix K.

## 3.1 Efficient RAG Module

$E^2$-RAG employs an encoder-decoder module for efficient RAG. The encoder pre-processes document chunks into compressed KV caches offline, which are then stored in a database. During online inference, the module retrieves the top-k KV caches from the database and feeds them to the decoder. The decoder generates responses based on these compressed KV caches. Our training process consists of two stages: pretraining and fine-tuning. Pretraining focuses on learning compression capabilities, while fine-tuning focuses on the question-answering task.

**Pretraining.** In this stage, we pretrain the efficient RAG module to acquire text compression capabilities. We use a frozen LLM, denoted as $\Theta_{\text{LLM}}$, with trainable LoRA $\Theta_{\text{LoRA}}^{\text{Encoder}}$ as the encoder, and the same frozen LLM as the decoder. Consider the input text tokens as $X$. After embedding, they are concatenated with trainable memory embedding $M = (m_1, m_2, \cdots, m_n)$, and $m_i \in \mathbb{R}^d$, where $d$ is the hidden size of the LLM. The outputs are the KV caches of $M$, $K = [k_1, k_2, \cdots, k_l]$, and $k_i \in \mathbb{R}^{h \times n \times d_h}$ is the "key" of $i$-th attention, where $l$ is the number of layers of the LLM, $h$ is the number of attention heads, and $d_h$ is the dimension of attention heads. $V = [v_1, v_2, \cdots, v_l]$, and $v_i \in \mathbb{R}^{h \times n \times d_h}$ is the "value" of $i$-th attention. Formally:

$$E = \mathbb{1}_X W_E \tag{1}$$

$$(K, V) = \text{encoder}([E, M]; \Theta_{\text{LLM}}, \Theta_{\text{LoRA}}^{\text{Encoder}}) \tag{2}$$

where $\mathbb{1}_X$ is the one-hot vector of $X$, $W_E$ is the embedding parameter of the LLM, and $E \in \mathbb{R}^{T \times d}$ is the embedding of $X$. KV cache $(K, V)$ is then passed to the frozen decoder to reconstruct the original text $X$. The reconstruction process minimizes the cross-entropy loss to train the LoRA $\Theta_{\text{LoRA}}^{\text{Encoder}}$:

$$\mathcal{L} = -\frac{1}{T} \sum_{t=1}^{T} \log P(x_t \mid x_{<t}; K, V; \Theta_{\text{LLM}}, \Theta_{\text{LoRA}}^{\text{Encoder}}) \tag{3}$$

**Fine-tuning.** For fine-tuning, we adapt the pretrained LoRA $\Theta_{\text{LoRA}}^{\text{Encoder}}$ to perform question answering based on relevant chunks $\{D_1, \cdots, D_k\}$. For each chunk $D_i$, we feed it to the encoder to obtain its KV cache, denoted as $(K_i, V_i)$. We then concatenate these KV caches, resulting in a combined KV cache denoted as $(K_{1:k}, V_{1:k})$. Let the question be $Q$ and the answer be $A$. The loss for the fine-tuning stage can be expressed as:

$$\mathcal{L} = -\frac{1}{T} \sum_{t=1}^{T} \log P(A \mid Q; K_{1:k}, V_{1:k}; \Theta_{\text{LLM}}, \Theta_{\text{LoRA}}^{\text{Encoder}}) \tag{4}$$

It is important to note that, to prevent the position embedding of the "key" from becoming disordered (Lu et al., 2024; Sun et al., 2024), the position embedding of the "key" must be removed when using the encoder to process chunks offline. When feeding them into the decoder, the position encoding should be reapplied according to the sorting. Details on repositioning can be found in Appendix C.

## 3.2 Editor Module

When updating the database is necessary, $E^2$-RAG first retrieves KV caches related to the new knowledge. The editor module then takes the new knowledge tokens as inputs and produces the offset KV for **INSERT**, **DELETE**, and **UPDATE** operations. Although these operations share the same architecture, their learning objectives differ.

**The Architecture for the Editor.** Our editor is a frozen LLM with trainable LoRA, designed to append, delete, and modify by processing the new knowledge $I$ into the offset KV cache $(\Delta K, \Delta V)$, which represents the updates to the KV cache. Formally:

$$\tilde{E} = \mathbb{1}_I W_E \tag{5}$$

$$(\Delta K, \Delta V) = \text{editor}([\tilde{E}, C]; \Theta_{\text{LLM}}, \Theta_{\text{LoRA}}^{\text{Editor}}) \tag{6}$$

where $C \in \mathbb{R}^{c \times d}$, where $c \ll n$, represents the trainable editing embedding, and $(\Delta K, \Delta V)$ denotes the KV cache of $C$. The updated KV cache is obtained by concatenating the modifications to the original cache:

$$[\tilde{K}, \Delta K] \rightarrow K \tag{7}$$

$$[\tilde{V}, \Delta V] \rightarrow V \tag{8}$$

To ensure that the edited KV caches can be used alongside other KV caches for multi-chunk inference, we propose maintaining a queue of KV caches to store the K and V values from previous batches. During each batch, we randomly sample from this queue and mix the selected KV caches. The detailed algorithm is presented in Appendix C.2.

The editor is also trained to minimize the cross-entropy loss:

$$\mathcal{L} = -\frac{1}{T} \sum_{t=1}^{T} \log P(A \mid Q; K, V; \Theta_{\text{LLM}}, \Theta_{\text{LoRA}}^{\text{Editor}}) \tag{9}$$

**Different Objectives for INSERT, DELETE, UPDATE.** Due to their distinct goals, all three operations—despite using the same editor architecture—have significantly different training objectives.

**INSERT**: This operation incorporates new knowledge. For a given question (related to the old or new knowledge), the decoder is first prefilled with the updated KV caches $([\tilde{K}, \Delta K], [\tilde{V}, \Delta V])$. The query $Q_{insert}$ of this question then attends to both the original knowledge in the caches and the newly appended knowledge. The attention mechanism computes the attention scores as follows:

$$[\tilde{A}, \Delta A] = \text{softmax}\left(\frac{Q_{insert} \cdot [\tilde{K}^\top, \Delta K^\top]}{\sqrt{d_k}}\right) \tag{10}$$

The resulting attention map $[\tilde{A}, \Delta A]$ assigns weights to the keys for both the original and new knowledge, capturing their relevance to the query.

**DELETE**: In contrast to **INSERT**, the **DELETE** operation applies a "mask" to suppress specific segments of old knowledge. When a question $Q_{\text{delete}}$ relates to the knowledge that needs to be removed, the new $\Delta V$ acts as a "mask" through the attention mechanism, effectively erasing the corresponding content in $\tilde{V}$:

$$O = (\tilde{A}\tilde{V} + \Delta A \Delta V) W_O^T \tag{11}$$

where, $O$ is the output of the attention mechanism and $W_O$ is the parameter of the output projection of the attention mechanism.

**UPDATE**: This operation combines the functionalities of **DELETE** and **INSERT**, as it involves both removing outdated knowledge and introducing new information. However, unlike **INSERT**, the **DELETE** and **UPDATE** operations may lead to *knowledge conflicts*. We discuss this in Section 6.4.

## 4 Editing Dataset Construction

For simplicity, we select documents with a length bounded by 128 tokens (the length of a single chunk) from five document QA datasets (Welbl et al., 2017; Yang et al., 2018; Dua et al., 2019; Rajpurkar, 2016; Stelmakh et al., 2022). We first split each document into a sequence of sentences: $\{s_1, \cdots, s_n\}$. For each document, we create a series of questions, where sentence $s_i$ contains the knowledge necessary to answer the question $Q$.

**INSERT.** We select one sentence $s_j$ from the document $\{s_1, \cdots, s_n\}$ as the new knowledge $I$, while the remaining sentences $\{s_1, \cdots, s_{j-1}, s_{j+1}, \cdots, s_n\}$ serve as the old context. When $i = j$, the new knowledge $I$ is necessary for answering $Q$. Conversely, when $i \neq j$, $Q$ can only be answered using the old context. For both cases, we provide the new document to Qwen2.5-7B (Yang et al., 2024) to generate responses to the questions.

**DELETE.** In contrast to **INSERT**, $s_j$ represents the sentence to be deleted, and the full set $\{s_1, \cdots, s_n\}$ constitutes the old full document. When $i = j$, the sentence necessary for answering $Q$ is deleted. Therefore, we use Qwen2.5-7B to input the remaining context $\{s_1, \cdots, s_{j-1}, s_{j+1}, \cdots, s_n\}$ along with the question to obtain a response indicating refusal to answer. For $i \neq j$, we generate a normal response, similar to the **INSERT** operation.

**UPDATE.** Similar to **DELETE**, the full set $\{s_1, \cdots, s_n\}$ serves as the old context. We use Qwen2.5-7B to generate new knowledge $\hat{s}_j$ that conflicts with $s_j$ and the corresponding question $\hat{Q}$. When $i = j$, the question $\hat{Q}$ is based on $\hat{s}_j$; when $i \neq j$, the questions are based on the old knowledge. The former requires the new knowledge $\hat{s}_j$; otherwise, the answer would be incorrect.

The data construction pipeline and examples of **INSERT**, **DELETE**, and **UPDATE** are provided in Appendix D. In the training set, both $i = j$ and $i \neq j$ account for 50%, ensuring that the model learns to insert new knowledge while retaining the old knowledge. In the test set, these two cases are separated to evaluate the model's ability to independently retain old knowledge and acquire new knowledge.

# 5 Experiments

**Setup.** We construct the data following the methods introduced in Section 4. Particularly, we use three datasets of HotpotQA (Yang et al., 2018), ASQA (Stelmakh et al., 2022) and Drop (Dua et al., 2019) for both training and testing, with two extra test sets of SciQ (Welbl et al., 2017) and SQuAD (Rajpurkar, 2016) for out-of-distribution (OOD) evaluation. We use Llama3.2-3B, Llama3.1-8B and Qwen2.5-7B (We present the results in Appendix H.) to train our models (Dubey et al., 2024; Yang et al., 2024). We train three separate editors for the three different types of editing operations. Training details, including hyperparameters, are provided in Appendix E. To ensure reliable evaluation, we adopt the Match metric, which checks whether the generated answer contains the golden answer as an exact match (Rau et al., 2024). This is particularly relevant since both the decoder in our method and the baseline standard RAG are frozen LLMs.

**Baselines.** We introduce two baselines as introduced in Section 2, standard RAG that directly updates documents in text form and our efficient RAG module that re-computes updated KV caches from edited text. As the two baselines require the edited full context, we come up with two setups.

**(1) LLM to Edit Text** In this baseline, we use an LLM to edit the old chunk based on the new knowledge $I$ and then answer the question. The LLM generates an updated context by inserting, deleting, or updating parts of the old chunk. While this approach can achieve reasonable performance, it lacks efficiency as it requires regenerating the entire context.

**(2) Golden Edit** For the **INSERT** operation, the golden edit baseline assumes access to the complete and up-to-date information, represented by the full set of sentences $\{s_1, \cdots, s_n\}$. This serves as an upper bound for performance, as it directly provides the model with the ideal context for answering the question $Q$. For **DELETE** and **UPDATE**, we annotate the editing results by humans of a subset from HotpotQA. The results can be referred to in Appendix F.

## 5.1 Results on Editing Efficiency

We assess the KV cache editing speed in Figure 2. When using the Llama3.1-8B model with a batch size of 16, our editor module achieves nearly a **40x** speedup compared to re-compute

KV caches (LLM to edit text + compress to new KV caches). This significantly reduces the time required for massive updates to the RAG database, as we perform editing at the KV cache level rather than using an LLM to edit text. We also include experiments on original document QA without editing in Appendix G, and our efficient RAG module also achieves **3x** speedup in response generation than standard RAG. We next evaluate the performance for three operations separately.

## 5.2 Results on INSERT Operation

Table 1: Results of **INSERT** operation on five document QA benchmarks and their averages. We use **"Old"** and **"New"** to indicate whether the knowledge involved in the question is added through the **INSERT** operation or is inherent to the old chunk. The results indicate that our method achieves nearly lossless in inserting. The best results are in **bold** and the second best are with underscore.

| Method | HotpotQA | | ASQA | | SciQ | | SQuAD | | Drop | | Avg. | |
|---|---|---|---|---|---|---|---|---|---|---|---|---|
| | Old (↑) | New (↑) | Old (↑) | New (↑) | Old (↑) | New (↑) | Old (↑) | New (↑) | Old (↑) | New (↑) | Old (↑) | New (↑) |
| **Standard RAG 3B** | | | | | | | | | | | | |
| golden edit | 77.54 | 80.91 | 76.53 | 78.53 | **89.29** | **89.86** | 85.86 | **87.22** | 56.62 | 60.13 | 77.17 | 79.33 |
| LLM to edit text | 73.19 | 74.55 | 74.18 | 75.46 | 88.93 | 89.49 | 83.93 | 85.42 | 56.42 | 58.23 | 75.33 | 76.63 |
| **E²-RAG w/o Editor (Efficient RAG 3B)** | | | | | | | | | | | | |
| golden edit | **81.16** | 79.09 | 76.53 | 79.14 | 86.79 | 85.51 | 78.57 | 80.74 | **74.75** | 71.88 | **79.56** | 79.27 |
| LLM to edit text | 76.81 | 72.73 | 76.53 | 75.46 | 84.64 | 82.61 | 77.71 | 76.69 | 69.65 | 58.54 | 77.09 | 73.11 |
| **E²-RAG 3B** | 78.99 | **87.27** | **77.00** | **85.89** | 82.50 | 86.59 | 75.43 | 82.90 | 74.54 | **79.11** | 77.69 | **84.33** |
| **Standard RAG 8B** | | | | | | | | | | | | |
| golden edit | 76.81 | 77.27 | 74.65 | 75.46 | 88.93 | **90.58** | 89.21 | 89.92 | 80.65 | 79.50 | 82.05 | 82.55 |
| LLM to edit text | 74.64 | 75.45 | 72.77 | 72.39 | 88.21 | 86.96 | 88.64 | 88.66 | 78.41 | 79.11 | 80.53 | 80.46 |
| **E²-RAG w/o Editor (Efficient RAG 8B)** | | | | | | | | | | | | |
| golden edit | 82.61 | **82.73** | **85.45** | **87.50** | **91.43** | 89.13 | 85.79 | 87.04 | 78.57 | 78.52 | **84.77** | **84.98** |
| LLM to edit text | **84.78** | 81.82 | 82.16 | 79.75 | 90.00 | 86.59 | 84.07 | 82.18 | 76.37 | 71.20 | 83.48 | 80.31 |
| **E²-RAG 8B** | 83.33 | 81.82 | 80.28 | 85.28 | 87.50 | 87.68 | 82.79 | 83.53 | **82.28** | **81.33** | 83.24 | 83.93 |

We report our results on the **INSERT** operation in Table 1. At both the Llama 3B and 8B sizes, our E²-RAG achieves first or second place on most benchmarks. Notably, the average score on "New" is **84.33** (first) in the 3B setting and 83.93 (second) in the 8B setting. The results on "Old" are slightly lower. We hypothesize that the KV cache for new knowledge is added after the original KV caches, bringing it closer to the query tokens, which causes the model to direct more attention toward the new knowledge. Among five benchmarks, we do not observe significant differences, indicating that our

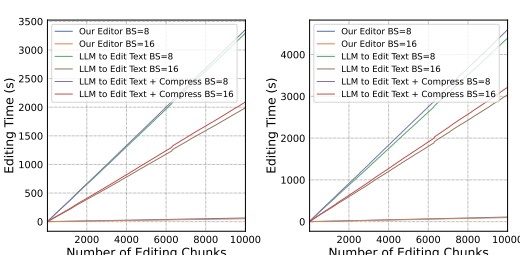

(a) Editing speed of 3B  (b) Editing speed of 8B

Figure 2: Editing Time for 3B/8B LLMs

approach can also adapt to the two OOD benchmarks, SciQ and SQuAD. We find that Efficient RAG sometimes outperforms standard RAG on certain benchmarks. One possible reason for this is the vulnerability of the current standard RAG system when presented with irrelevant or misleading information. In contrast, compressing the context into KV caches effectively extracts and simplifies information, omitting less important details, which enhances the model's retrieval ability (Cheng et al., 2024).

## 5.3 Results on DELETE Operation

The results for the **DELETE** operation are presented in Table 2. Unlike **INSERT**, "Rm" represents questions that require the removed knowledge. Therefore, a lower "Rm" value indicates a higher degree of knowledge removal. From the table, we observe that both the 3B and 8B models rank first or second in most benchmarks. For example, the average for "Rm" in the 3B model is 28.86 (second), while the average for the 8B model is **21.68** (first). However, the

Table 2: Results of **DELETE** operation on five document QA benchmarks and their averages. During the experiments, we apply the **DELETE** operation on the context in QA as described in the paper. We use **"Old"** to indicate that the QA only involves knowledge that is not deleted. **"Rm"** denotes QA involving deleted knowledge, where **lower scores indicate more effective deletion**.

| Method | HotpotQA | | ASQA | | SciQ | | SQuAD | | Drop | | Avg. | |
|---|---|---|---|---|---|---|---|---|---|---|---|---|
| | Old (↑) | Rm (↓) | Old (↑) | Rm (↓) | Old (↑) | Rm (↓) | Old (↑) | Rm (↓) | Old (↑) | Rm (↓) | Old (↑) | Rm (↓) |
| **Standard RAG 3B** | | | | | | | | | | | | |
| LLM to edit text | 70.27 | 33.87 | 71.62 | 20.69 | **87.24** | 48.21 | **78.92** | 29.75 | 53.56 | 28.33 | 72.32 | 32.17 |
| **E²-RAG w/o Editor (Efficient RAG 3B)** | | | | | | | | | | | | |
| LLM to edit text | **83.11** | 25.81 | **78.38** | **16.38** | 86.21 | **31.25** | 77.54 | 30.66 | **69.75** | 33.33 | **79.00** | 27.49 |
| E²-RAG 3B | 78.38 | 33.87 | 71.17 | 17.24 | 84.83 | 39.29 | 76.23 | 30.57 | 68.86 | 23.33 | 75.89 | 28.86 |
| **Standard RAG 8B** | | | | | | | | | | | | |
| LLM to edit text | 70.27 | 27.42 | 70.72 | 13.79 | 87.24 | 34.82 | 88.74 | 27.30 | 73.13 | 6.67 | 78.02 | 22.00 |
| **E²-RAG w/o Editor (Efficient RAG 8B)** | | | | | | | | | | | | |
| LLM to edit text | 87.16 | 32.26 | **86.94** | 15.52 | **88.28** | 25.89 | **87.77** | 29.30 | **78.29** | 23.33 | **85.69** | 25.26 |
| E²-RAG 8B | 78.38 | 30.65 | 79.28 | **6.03** | 85.17 | **21.43** | 80.17 | **25.30** | 72.24 | 25.00 | 79.05 | **21.68** |

performance on "Old" is slightly lower, suggesting that direct concatenation of KV caches may not be the optimal solution to retain old knowledge. We leave this for future study.

## 5.4 Results on **UPDATE** Operation

Table 3: Results of the **UPDATE** operation on five benchmarks. **"Rp"** (replace) denotes QA involving the updated knowledge.

| Method | HotpotQA | | ASQA | | SciQ | | SQuAD | | Drop | | Avg. | |
|---|---|---|---|---|---|---|---|---|---|---|---|---|
| | Old (↑) | Rp (↑) | Old (↑) | Rp (↑) | Old (↑) | Rp (↑) | Old (↑) | Rp (↑) | Old (↑) | Rp (↑) | Old (↑) | Rp (↑) |
| **Standard RAG 3B** | | | | | | | | | | | | |
| LLM to edit text | 76.27 | 57.20 | 64.17 | 66.25 | 52.60 | 61.81 | **60.89** | 63.81 | 49.30 | 72.30 | 60.65 | 64.27 |
| **E²-RAG w/o Editor (Efficient RAG 3B)** | | | | | | | | | | | | |
| LLM to edit text | 76.27 | 61.86 | 65.69 | 71.94 | 53.64 | 64.19 | 57.96 | 72.38 | 51.17 | 74.51 | 60.95 | 68.98 |
| E²-RAG 3B | **84.75** | **83.90** | **68.47** | **79.86** | **55.13** | **78.90** | 60.79 | **79.74** | 48.83 | 71.36 | **63.59** | **78.75** |
| **Standard RAG 8B** | | | | | | | | | | | | |
| LLM to edit text | 84.32 | 70.34 | 67.78 | 71.39 | 57.80 | 64.78 | 66.23 | 73.39 | 64.32 | 87.79 | 68.09 | 73.54 |
| **E²-RAG w/o Editor (Efficient RAG 8B)** | | | | | | | | | | | | |
| LLM to edit text | 84.32 | 74.15 | 70.42 | 75.14 | 60.77 | 68.35 | 66.63 | 74.40 | 63.38 | 79.32 | 69.10 | 74.27 |
| E²-RAG 8B | **85.17** | **84.75** | 68.19 | **84.86** | 57.50 | **80.53** | 62.00 | **83.27** | **75.12** | 81.83 | **69.60** | **83.05** |

For the **UPDATE** operation, we present the results in Table 3. As discussed earlier, **UPDATE** is a combination of **INSERT** and **DELETE**, requiring both the removal of outdated knowledge and the insertion of new knowledge. This operation is inherently more challenging as it introduces potential knowledge conflicts. Notably, E²-RAG performs well on both "Old" and "New" (which require updated knowledge) for both the 3B and 8B Llama models.

# 6 Ablation and Discussion

To provide further insights into knowledge updating for RAG, we conduct a series of ablation studies on multi-turn editing and multi-chunk capability for **INSERT**. Additionally, we discuss knowledge conflicts that arise during the editing process for **DELETE** and **UPDATE**.

## 6.1 Multi-turn Editing

In practical applications, it is often necessary to update a specific chunk multiple times. Therefore, we conduct experiments on multi-turn editing using data from the **INSERT** operation on both the 3B and 8B models. We present the average results in Appendix I. Our findings indicate that as the number of editing iterations increases, both the old knowledge and the newly inserted knowledge from the first round gradually fade. Consequently,

we recommend reconstructing the chunk from the original text after a maximum of three editing operations to maintain information integrity.

## 6.2 Multi-chunk Ability with Edited Chunks

Real-world applications often require models to retrieve and utilize information from multiple chunks to answer questions. As mentioned in Section 3, our editor is trained with a chunk queue to enhance its capability of handling multiple chunks effectively. To systematically evaluate this ability, we conduct experiments by introducing varying amounts of noisy chunks into the test set, ranging from 0 to 9. This evaluation covers scenarios where the total number of chunks varies from 1 to 10, corresponding to the Top-K values used in common RAG settings (Mao et al., 2024). This setup allows

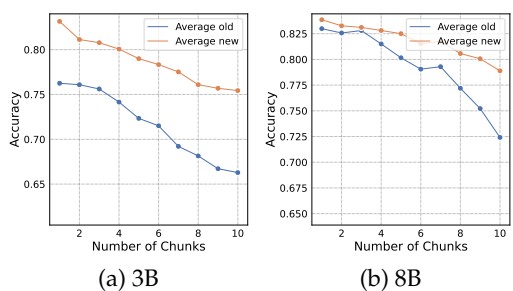

(a) 3B          (b) 8B

Figure 3: Multi-chunk Ability on (a) 3B size and (b) 8B size.

us to assess how well the editor maintains knowledge consistency while retrieving relevant information in multi-chunk settings. We present the average results across five benchmarks in Figure 3, with detailed evaluation results available in Appendix J. The results indicate that as the number of chunks increases, both "Old" and "New" knowledge performance slightly decreases. This decline occurs because the increase in the number of chunks leads to more frequent errors in information retrieval. Specifically, as more chunks are involved, the model faces greater challenges in accurately retrieving relevant information. By comparing the results of the 3B and 8B models, we observe that the multi-chunk ability of the 8B model is significantly better than that of the 3B model, suggesting that multi-chunk capability is correlated with model size.

## 6.3 Robustness on Insufficient Updating

A potential issue with updating only the top-$k$ relevant entries in the database is that other outdated entries (the remaining $n - k$) are left unchanged. These outdated entries may degrade model performance. We conduct an experiment to investigate the robustness of our method in this scenario. Specifically, we select approximately 100 questions from the test set of Squad for the three operations and duplicate the associated information chunk five times. Subsequently, the ratio of edited (updated) chunks could be varied systematically from 0.2 (1 out of 5) to 1.0 (all 5 out of 5). This would allow for an examination of model performance under varying degrees of information conflict. We provide the results for Llama-3.2 3B and Llama-3.1 8B in Table 4.

Table 4: Performance under different ratios of edited knowledge chunks. The results demonstrate a reasonable robustness to incomplete updates.

| Model Size | Operation | Ratio of Edited Chunks | | | | |
|---|---|---|---|---|---|---|
| | | 0.2 | 0.4 | 0.6 | 0.8 | 1.0 |
| 3B | INSERT (↑) | 0.40 | 0.58 | 0.61 | 0.60 | 0.63 |
| | DELETE (↓) | 0.31 | 0.34 | 0.22 | 0.19 | 0.20 |
| | UPDATE (↑) | 0.75 | 0.77 | 0.82 | 0.82 | 0.82 |
| 8B | INSERT (↑) | 0.69 | 0.71 | 0.75 | 0.81 | 0.81 |
| | DELETE (↓) | 0.38 | 0.39 | 0.25 | 0.16 | 0.14 |
| | UPDATE (↑) | 0.79 | 0.81 | 0.82 | 0.86 | 0.86 |

Firstly, all the models and operations demonstrate a similar trend where performance slightly drops when the ratio of edited chunks is reduced from 1.0 to 0.2. This is a reasonable

result because when the number of edited chunks decreases, the model can be influenced by the unedited chunks, which may confuse the model and lead to incorrect answers. However, to our surprise, our method also shows a certain level of robustness in this scenario. Especially in the UPDATE operation, even when only 20% of the chunks are updated, the model can still notice the change and answer correctly. For instance, the performance of the 3B model remains at 0.75 when the ratio of edited chunks is only 0.2, compared to 0.82 when the ratio is 1.0.

We acknowledge that updating by selecting the top-k relevant outdated entries is indeed a naive method, as it can lead to the issue of incomplete updates, as you mentioned. However, since our work is the first to explore how to efficiently keep the RAG database up-to-date, we believe there will be future follow-up work to better address this challenge.

### 6.4 Knowledge Conflicts

The **DELETE** and **UPDATE** operations may result in knowledge conflicts; however, our model still performs decently. Therefore, it is important to investigate the underlying reasons. For the **DELETE** operation, we select two types of questions: one related to old knowledge that can be answered correctly ("Old"), and another solely related to the removed knowledge ("Rm"), which cannot be answered correctly by the model. We plot the attention map of the question over the edited chunk in Figure 4. We observe that when the question pertains to the removed knowledge, the attention to $\Delta K$ significantly increases. We conduct the same experiment for the **UPDATE** operation, and the results are

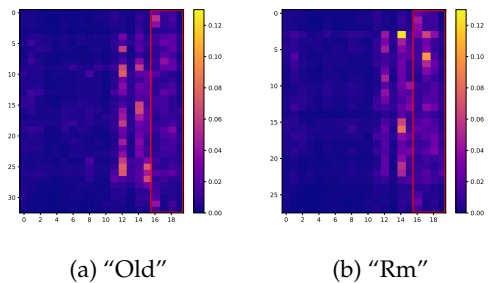

(a) "Old"        (b) "Rm"

Figure 4: Attention map of the **DELETE** operation, (a) The question is only related to the old knowledge. (b) The question is only related to the removed knowledge. The red box in both sub-figure means the attention of $\Delta K$.

consistent with those of the **DELETE** operation. We hypothesize that in **DELETE**, when the question requires information from offset KV caches, $\Delta V$ acts as a "mask" to nullify certain portions of the old knowledge's attention "value", while in **UPDATE**, it functions to replace the old knowledge.

## 7 Conclusion and Future Work

This work addresses the challenge of efficient editing in compressed KV caches of RAG. We propose E$^2$-RAG, the first editable efficient RAG architecture that balances both editing and inference efficiency. To evaluate our approach, we construct three datasets specifically designed to assess the basic operations of **INSERT**, **DELETE**, and **UPDATE**. Our comprehensive experiments demonstrate the editability and efficiency of the proposed methods, along with further ablations and discussions that provide valuable insights into the challenges associated with this problem. Our findings highlight the potential for further advancements in the editability and efficiency of RAG.

## Acknowledgment

This work was supported by Huawei, the Shenzhen Science and Technology Program (JCYJ20220818103001002), Shenzhen Doctoral Startup Funding (RCBS20221008093330065), Tianyuan Fund for Mathematics of National Natural Science Foundation of China (NSFC) (12326608), Shenzhen Science and Technology Program (Shenzhen Key Laboratory Grant No. ZDSYS20230626091302006), and Shenzhen Stability Science Program 2023, Shenzhen Key Lab of Multi-Modal Cognitive Computing.

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

## A  Related Work

As the context length increases (Li et al., 2024a), the inference cost of the RAG model increases significantly (Ram et al., 2023; Salemi & Zamani, 2024). To enhance the efficiency of RAG, efficient RAG methods have emerged, aiming to reduce time overhead by optimizing the entire retrieval-generation workflow.

In earlier studies, numerous approaches have been developed to enhance the efficiency of RAG. The reduction of input context as a natural method was the first to be validated. Initially, various text summarization techniques (Xu et al., 2023; Jiang et al., 2024; Edge et al., 2024; Chan et al., 2024) were introduced to minimize information loss. Knowledge distillation and filtering techniques are employed in (Shi et al., 2024; Liu et al., 2023; Sarthi et al., 2024) and (Jung et al., 2024) to compress lengthy texts and preserve essential knowledge. Furthermore, some works (Shahout et al., 2024a; Ray et al., 2024) controlled the effective length of the input by addressing information redundancy. Although these

approaches effectively reduced context length, they resulted in semantic loss and poor scalability.

Recent studies (Leviathan et al., 2023; Shahout et al., 2024b; Miao et al., 2023; Nagel et al., 2019; Aminabadi et al., 2022) have aimed to enhance efficiency by focusing on the large models themselves. We emphasize works that compress the context into LLM intermediate components such as embeddings (Mu et al., 2024; Chevalier et al., 2023; Ge et al., 2023; Li et al., 2024b) and KV caches (Lu et al., 2024; Sun et al., 2024). Although these methods are effective and incur minimal loss, they are challenging to update. In contrast to prior research, our focus is on editable efficient RAG.

Previous works (De Cao et al., 2021; Dong et al., 2022; Meng et al., 2022) on knowledge editing focus on inserting or updating knowledge in LLMs. And many works (Li et al., 2021; Tong et al., 2022; Yao et al., 2023; Ullah et al., 2021) on machine unlearning aim to remove specific knowledge from LLMs. Our work applies three types of knowledge operations—**INSERT**, **DELETE**, and **UPDATE** —to the RAG database, ensuring that it remains up-to-date.

# B   Details of GPU hours in editing

We choose Llama-3.1-8B (Dubey et al., 2024) as the editor, utilizing vllm (Kwon et al., 2023) and flash-attention-2 (Dao, 2024) to accelerate inference with bfloat16 precision. The token throughput is approximately 1300 tokens/s. Each text chunk consists of 128 tokens, and the total sum of input and output tokens is 256. Therefore, the number of tokens to be processed is $T = 0.6 \times 10^6 \times 256$, and the required time is $S = \frac{T}{1300 \times 60 \times 60} = 32.8$ hours. So, the total A100 GPU time required is 32.8 hours.

# C   Details of Reposition and Chunks Queue

## C.1   Reposition

One challenge in using KV caches as chunks for RAG is the confusion caused by the position embedding of the "key". To address this, we remove the position embedding of the "key" when storing it in the database and reassign the position embedding to the key before pre-filling. We refer to this process as "Reposition".

Consider the "query" at position $m$ as $Q_m \in \mathbb{R}^d$, and the "query" with position embedding as $\mathcal{Q}_m$. RoPE adds position encoding to $Q_m$ through the following formula:

$$\mathcal{Q}_m = Q_m \otimes \cos_{\Theta;m} + \text{Rotary Half}(Q_m) \otimes \sin_{\Theta;m} \tag{12}$$

$$= \begin{pmatrix} q_0 \\ q_1 \\ q_2 \\ q_3 \\ \vdots \\ q_{d-2} \\ q_{d-1} \end{pmatrix} \otimes \begin{pmatrix} \cos m\theta_0 \\ \cos m\theta_0 \\ \cos m\theta_1 \\ \cos m\theta_1 \\ \vdots \\ \cos m\theta_{d/2-1} \\ \cos m\theta_{d/2-1} \end{pmatrix} + \begin{pmatrix} -q_1 \\ q_0 \\ -q_3 \\ q_2 \\ \vdots \\ -q_{d-1} \\ q_{d-2} \end{pmatrix} \otimes \begin{pmatrix} \sin m\theta_0 \\ \sin m\theta_0 \\ \sin m\theta_1 \\ \sin m\theta_1 \\ \vdots \\ \sin m\theta_{d/2-1} \\ \sin m\theta_{d/2-1} \end{pmatrix} \tag{13}$$

For the "key" at position $m$, $K_m$, position encoding is added in the same manner to obtain $\mathcal{K}_m$. However, instead of adding the position encoding, our goal is to recover $K_m$ given $m$ and $\mathcal{K}_m$. First, we have the formula for RoPE:

$$\mathcal{K}_m = K_m \otimes \cos_{\Theta;m} + \text{Rotary Half}(K_m) \otimes \sin_{\Theta;m} \tag{14}$$

$$\tag{15}$$

Additionally, we have the basic properties of Rotary and sine-cosine functions:

$$K_m = -\text{Rotary Half}(\text{Rotary Half}(K_m)) \tag{16}$$

$$1 = \sin^2_{\Theta;m} + \cos^2_{\Theta;m} \tag{17}$$

By solving the system of these three equations, we can obtain:

$$K_m = \mathcal{K}_m \otimes \cos_{\Theta;m} - \text{Rotary Half}(\mathcal{K}_m) \otimes \sin_{\Theta;m} \tag{18}$$

In this way, we can remove the position encoding from the "key".

## C.2 Chunks Queue

---

**Algorithm 1** Chunks Queue In Editor Training

---

1: **Input:** Batch size $B$, queue size $q$, queue $\mathcal{Q}$
2: Initialize an empty queue $\mathcal{Q}$ with maximum size $q$
3: **for** each batch $b$ **do**
4:     Get the compressed KV $(\tilde{K}_b, \tilde{V}_b)$ and $(\Delta K, \Delta V)$ from editor, and update to $(K_b, V_b)$
5:     Randomly sample a subset $(K_{1:r}, V_{1:r}) \subseteq \mathcal{Q}$
6:     Construct multi-chunk KV caches and random sort: $\text{Random Sort}[(K_1, V_1), \ldots, (K_r, V_r), (K_b, V_b)]$
7:     Enqueue $(\tilde{K}_b, \tilde{V}_b)$ and $(K_b, V_b)$ into $\mathcal{Q}$
8:     **if** queue size $> q$ **then**
9:         Remove the oldest KV caches from $\mathcal{Q}$
10:     **end if**
11: **end for**

---

# D   Examples of Three Type of Editing Operations

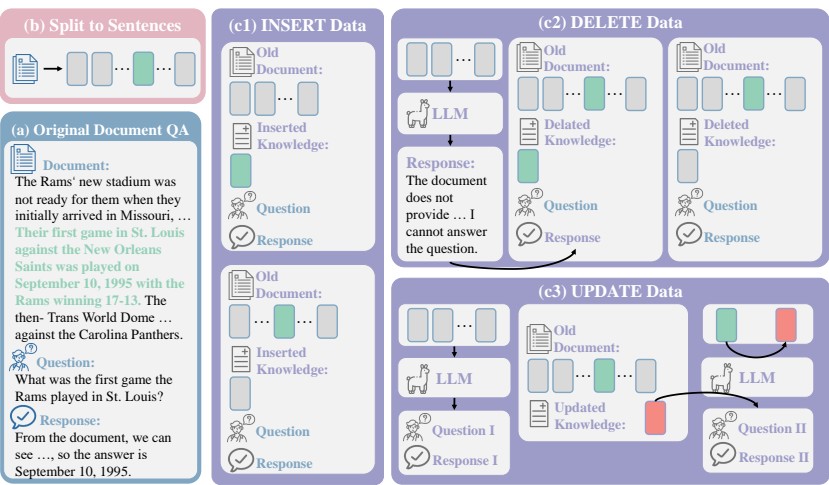

Figure 5: The pipeline of our data creation. (a) We use the original document QA datasets, and (b) we first split the context into sentences. (c1) For the **INSERT** operation, we randomly select one of the sentences as new knowledge and reuse the question from the original document QA. (c2) For the **DELETE** operation, when the removed knowledge is related to the question, we use an LLM to generate a response that refuses to answer the question. (c3) For the **UPDATE** operation, we only use the context from the document QA, and then create conflicting new knowledge using an LLM. On one hand, we use this new knowledge to generate a question and its corresponding response; on the other hand, we use other sentences to generate a question unrelated to the new knowledge and its response.

---

**An Example of Insert Operation**

Old Document:
Founded in 1934, the company owns and/or operates 30+ luxury hotels and two river cruise ships in six countries, primarily under its Oberoi Hotels & Resorts and Trident Hotels brands. The Oberoi family is an Indian family that is famous for its involvement in hotels, namely through The Oberoi Group. Below is an example of knowledge addition.
New Knowledge:
The Oberoi Group is a hotel company with its head office in Delhi.
Question:
The Oberoi family is part of a hotel company that has a head office in what city?
Answer:
Delhi

Figure 6: An example of the data of **INSERT**, the question is related to the new knowledge.

---

**An Example of Insert Operation**

Old Document:
Bizarre was published by Dennis Publishing, and was a sister publication to the Fortean Times. Fortean Times is a British monthly magazine devoted to the anomalous phenomena popularised by Charles Fort. Previously published by John Brown Publishing (from 1991 to 2001) and then I Feel Good Publishing (2001 to 2005), it is now published by Dennis Publishing Ltd.
New Knowledge:
Bizarre was a British alternative magazine published from 1997 to 2015.
Question:
Which publishing company has published Bizarre and a sister publication devoted to the anomalous phenomena popularised by Charles Fort?
Answer:
Dennis Publishing

Figure 7: An example of the data of **INSERT**, the question is related to the old knowledge.

---

**An Example of Delete Operation**

Old Document:
Jo Ann Terry-Grissom (born August 4, 1938 in Indianapolis, Indiana) is a retired female hurdler from the United States, who represented her native country at two consecutive Summer Olympics, starting in 1960. Affiliated with the Tennessee State University she won the 80 m hurdles event at the 1963 Pan American Games. The 4th Pan American Games were held from April 20 to May 5, 1963, in Sao Paulo, Brazil.
Removed Knowledge:
The 4th Pan American Games were held from April 20 to May 5, 1963, in Sao Paulo, Brazil.
Question:
Jo Ann Terry won the 80m hurdles event at what Sao Paulo-based event from 1963?
Answer:
Pan American Games

Figure 8: An example of the data of **DELETE**, the question is related to the removed knowledge.

---

**An Example of Delete Operation**

Old Document:
Nicholas Sposato serves on Chicago City Council as alderman of the 38th Ward of the City of Chicago on the city's Northwest Side. Sposato was elected in 2011 in an election against incumbent alderman John Rice, who was endorsed by then Mayor-elect Rahm Emanuel. Rahm Israel Emanuel ( ; born November 29, 1959) is an American politician who is the 44th and current mayor of Chicago. A member of the Democratic Party, Emanuel was elected in 2011. He was re-elected on April 7, 2015.
Removed Knowledge:
A member of the Democratic Party, Emanuel was elected in 2011.
Question:
What side of town is the Chicago Mayer-endorsed, 38th Ward of the City of Chicago alderman serving?
Answer:
Northwest Side

---

Figure 9: An example of the data of **DELETE**, the question is related to the old knowledge.

---

**An Example of Update Operation**

Old Document:
What Happened to Jones is a 1926 silent film comedy directed by William A. Seiter and starring Reginald Denny. It was produced and distributed by Universal Pictures. The film is taken from an 1897 Broadway play, What Happened to Jones by George Broadhurst. Reginald Denny (born Reginald Leigh Dugmore, 20 November 1891, 16 June 1967) was an English stage, film and television actor as well as an aviator and UAV pioneer. He was once an amateur boxing champion of Great Britain.
Conflicting New Knowledge:
What Happened to Jones is a 1990 color film.
Question:
When was What Happened to Jones released?
Answer:
1990

---

Figure 10: An example of the data of **UPDATE**, the question is related to the conflicting new knowledge.

---

**An Example of Update Operation**

Old Document:
Double Take is a 2001 action comedy film starring Eddie Griffin and Orlando Jones. Double Take was inspired by the 1957 drama Across the Bridge, which was in turn based on a short story by Graham Greene; the supporting cast includes Edward Herrmann, Gary Grubbs, Garcelle Beauvais, and Daniel Roebuck. Gary Grubbs (born November 14, 1949) is an American actor.
Conflicting New Knowledge:
Gary Grubbs was born on January 1, 1980.
Question:
When was the film Double Take released?
Answer:
2001

---

Figure 11: An example of the data of **UPDATE**, the question is related to the old knowledge.

## E  Training Details

We use transformers (Wolf, 2019) to train our model, and the hyperparameters are shown in Table 5. In pretraining, we only use one chunk to train models to get the ability of compression. During finetuning, we use multi-chunk data to cultivate the model to retrieve from multiple chunks and answer the question. To train the editor, we use a chunk queue to cache the chunks from previous batches which we describe in Section 3.

Table 5: Training Hyperparameters

| | Pretraining | | Finetuning | | Editor training | |
|---|---|---|---|---|---|---|
| | 3B | 8B | 3B | 8B | 3B | 8B |
| LoRA rank | 64 | 64 | 64 | 64 | 64 | 64 |
| Learning rate | 1e-4 | 1e-4 | 1e-4 | 1e-4 | 1e-4 | 1e-4 |
| Batch size | 32 | 32 | 32 | 32 | 32 | 32 |
| Steps | 16000 | 16000 | 18000 | 12000 | 1600 | 1600 |
| Warm-up steps | 300 | 300 | 300 | 300 | 100 | 100 |
| memory tokens | 16 | 16 | 16 | 16 | 16 | 16 |
| edit tokens | - | - | - | - | 4 | 4 |
| chunk size | 128 | 128 | 128 | 128 | 128 | 128 |
| Max chunks | 1 | 1 | 10 | 10 | 10 | 10 |
| Queue size | - | - | - | - | 20 | 20 |

## F   Results of Golden Editing by Human

Table 6: Results of HotpotQA of **UPDATE** and **DELETE** with golden editing by human.

| Method | UPDATE | | DELETE | |
|---|---|---|---|---|
| | Old ($\uparrow$) | Rp ($\uparrow$) | Old ($\uparrow$) | Rm ($\downarrow$) |
| **Standard RAG 3B** | | | | |
| golden edit | 85.59 | 80.93 | 69.59 | 24.19 |
| LLM to edit text | 76.27 | 57.20 | 70.27 | 33.87 |
| **E$^2$-RAG w/o Editor (Efficient RAG 3B)** | | | | |
| golden edit | 86.86 | 74.15 | 81.76 | 19.35 |
| LLM to edit text | 76.27 | 61.86 | 83.11 | 25.81 |
| **E$^2$-RAG 3B** | 84.75 | 83.90 | 78.38 | 33.87 |
| **Standard RAG 8B** | | | | |
| golden edit | 85.59 | 76.27 | 74.32 | 17.74 |
| LLM to edit text | 84.32 | 70.34 | 70.27 | 27.42 |
| **E$^2$-RAG w/o Editor (Efficient RAG 8B)** | | | | |
| golden edit | 88.98 | 82.63 | 89.86 | 16.13 |
| LLM to edit text | 84.32 | 74.15 | 87.16 | 32.26 |
| **E$^2$-RAG 8B** | 85.17 | 84.75 | 78.38 | 30.65 |

## G   Results on RAG benchmarks

We use five document QA benchmarks (Welbl et al., 2017; Yang et al., 2018; Dua et al., 2019; Rajpurkar, 2016; Stelmakh et al., 2022) and two long-form QA benchmarks (Lin et al., 2021; Rosenthal et al., 2025) to comprehensively demonstrate the RAG performance of E$^2$-RAG. To evaluate the efficiency of QA, we measure the Time To First Token (TTFT) and overall latency under varying context lengths and batch sizes.

In Table 7, we present the performance of standard RAG and E$^2$-RAG on various document QA benchmarks. E$^2$-RAG shows performance comparable to standard RAG on these tasks and even outperforms standard RAG on some benchmarks. For the long-form QA benchmarks in Table 8, E$^2$-RAG outperforms standard RAG in most cases. This improvement is likely due to the addition of retrieval and reasoning steps during the finetuning stage. For efficiency, first, we evaluate the TTFT across different batch sizes and varying context lengths, with the results shown in Figure 12b and Figure 12a. We observe that the TTFT of E$^2$-RAG remains nearly constant at a very low level. This is because E$^2$-RAG pre-processes the context into compressed KV caches offline. Second, we present the total inference latency

for generating different numbers of tokens at a fixed context length of 2560 across various batch sizes. As shown in Figure 12d, at the 8B model size and a batch size of 8, E$^2$-RAG achieves a 3x speedup compared to standard RAG. This speedup increases with larger batch sizes and model sizes. The improvement is attributed to the offline prefilling process and the significant compression of the context, which reduces the context length and accelerates the next token generation.

Table 7: Experimental results of **E$^2$-RAG** on five document QA benchmarks and their averages. The results show that **E$^2$-RAG** incurs slight loss but overall maintains an advantage in document QA.

| Model | Method | HotpotQA | ASQA | SciQ | SQuAD | Drop | Avg. |
|---|---|---|---|---|---|---|---|
| Llama 3.2 3B | w/o contexts | 21.92 | 38.37 | 60.9 | 18.28 | 16.22 | 31.14 |
| | standard RAG | 74.38 | 74 | 82.1 | 85.89 | 68.27 | 76.93 |
| | **E$^2$-RAG** | **74.96** | **79.14** | **85.6** | 78.33 | 63.06 | 76.22 |
| | Δ | +0.58 | +5.14 | +3.5 | -7.56 | -5.21 | -0.71 |
| Llama 3.1 8B | w/o contexts | 27.57 | 49.56 | 66.8 | 24.76 | 18.76 | 37.49 |
| | standard RAG | 75.88 | 79.93 | 89.5 | 89.55 | 74.68 | 81.91 |
| | **E$^2$-RAG** | **79.36** | **84.99** | 87.8 | 83.62 | 67.64 | 80.68 |
| | Δ | +3.48 | +5.06 | -1.7 | -5.93 | -7.04 | -1.23 |

Table 8: Experimental results of **E$^2$-RAG** on two long-form QA benchmarks. The results show that **E$^2$-RAG** has a certain advantage on long-form QA benchmarks.

| Size | Method | TruthfulQA | | ClapNQ | |
|---|---|---|---|---|---|
| | | F1 | R-L | F1 | R-L |
| 3B | w/o contexts | 9.6 | 10.09 | 9.17 | 8.25 |
| | standard RAG | 25.59 | 26.62 | 20.76 | 18.36 |
| | **E$^2$-RAG** | 26.16 | 26.75 | 19.73 | 18.71 |
| | Δ | +0.57 | +0.13 | -1.03 | +0.35 |
| 8B | w/o contexts | 9.66 | 9.86 | 9.38 | 8.35 |
| | standard RAG | 19.38 | 24.35 | 19.92 | 17.68 |
| | **E$^2$-RAG** | 26.39 | 27.09 | 20.23 | 19.09 |
| | Δ | +7.01 | +2.74 | +0.31 | +1.41 |

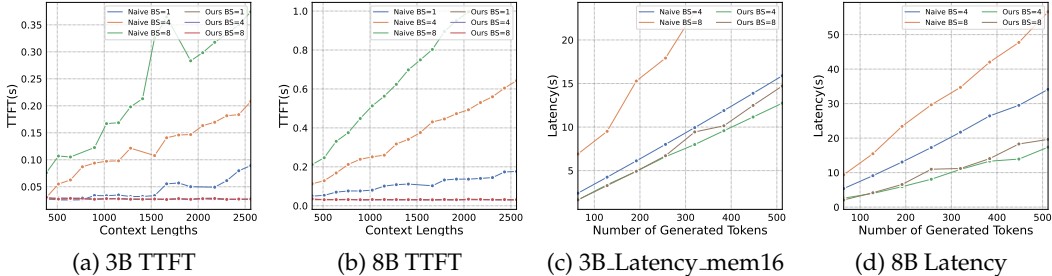

(a) 3B TTFT     (b) 8B TTFT     (c) 3B_Latency_mem16     (d) 8B Latency

Figure 12: The TTFT and Latency of E$^2$-RAG and standard RAG on both 3B and 8B.

# H   Results on Qwen

To further validate our conclusions, we conduct additional experiments on Qwen2.5-7B (Yang et al., 2024).

Table 9: Results of Qwen 2.5 7B on **INSERT** operation.

| Method | HotpotQA | | ASQA | | SciQ | | SQuAD | | Drop | | Avg. | |
|---|---|---|---|---|---|---|---|---|---|---|---|---|
| | Old (↑) | New (↑) | Old (↑) | New (↑) | Old (↑) | New (↑) | Old (↑) | New (↑) | Old (↑) | New (↑) | Old (↑) | New (↑) |
| **Standard RAG 7B** | | | | | | | | | | | | |
| golden context | 86.96 | 88.18 | 88.26 | 92.02 | 91.79 | 91.30 | 92.43 | 92.62 | 78.00 | 84.18 | 87.49 | 89.66 |
| LLM to edit text | 84.06 | 87.27 | 86.85 | 89.57 | 90.71 | 91.30 | 92.43 | 92.17 | 77.19 | 81.96 | 86.25 | 88.45 |
| **E$^2$-RAG w/o Editor (Efficient RAG 7B)** | | | | | | | | | | | | |
| golden context | 86.23 | 86.36 | 86.38 | 85.94 | 91.07 | 89.86 | 84.07 | 83.44 | 80.04 | 75.95 | 85.56 | 84.31 |
| LLM to edit text | 79.71 | 79.09 | 85.92 | 79.14 | 88.57 | 87.68 | 82.14 | 81.01 | 75.97 | 71.20 | 82.46 | 79.62 |
| **E$^2$-RAG 7B** | 84.06 | 82.73 | 83.10 | 83.44 | 89.29 | 89.49 | 82.00 | 81.55 | 82.48 | 79.43 | 84.19 | 83.33 |

Table 10: Results of Qwen2.5-7B on **DELETE** operation.

| Method | HotpotQA | | ASQA | | SciQ | | SQuAD | | Drop | | Avg. | |
|---|---|---|---|---|---|---|---|---|---|---|---|---|
| | Old (↑) | Rm (↓) | Old (↑) | Rm (↓) | Old (↑) | Rm (↓) | Old (↑) | Rm (↓) | Old (↑) | Rm (↓) | Old (↑) | Rm (↓) |
| **Standard RAG 7B** | | | | | | | | | | | | |
| LLM to edit text | 81.76 | 35.48 | 84.23 | 7.76 | 89.66 | 35.71 | 91.09 | 23.75 | 74.20 | 25.00 | 84.19 | 25.54 |
| **E$^2$-RAG w/o Editor (Efficient RAG 7B)** | | | | | | | | | | | | |
| LLM to edit text | 79.14 | 32.26 | 88.29 | 7.76 | 88.97 | 17.86 | 85.90 | 21.29 | 77.40 | 25.00 | 83.94 | 20.83 |
| **E$^2$-RAG 7B** | 75.00 | 30.65 | 84.88 | 12.07 | 85.52 | 20.16 | 85.47 | 19.74 | 75.45 | 23.33 | 81.26 | 21.19 |

Table 11: Results of Qwen 2.5 7B on **UPDATE** operation.

| Method | HotpotQA | | ASQA | | SciQ | | SQuAD | | Drop | | Avg. | |
|---|---|---|---|---|---|---|---|---|---|---|---|---|
| | Old (↑) | Rp (↑) | Old (↑) | Rp (↑) | Old (↑) | Rp (↑) | Old (↑) | Rp (↑) | Old (↑) | Rp (↑) | Old (↑) | Rp (↑) |
| **Standard RAG 7B** | | | | | | | | | | | | |
| LLM to edit text | 88.14 | 77.54 | 73.89 | 82.78 | 65.53 | 75.19 | 69.15 | 82.36 | 75.12 | 88.26 | 74.37 | 81.23 |
| **E$^2$-RAG w/o Editor (Efficient RAG 7B)** | | | | | | | | | | | | |
| LLM to edit text | 84.32 | 72.88 | 69.72 | 81.11 | 62.56 | 72.66 | 65.62 | 78.93 | 74.65 | 84.51 | 71.37 | 78.02 |
| **E$^2$-RAG 7B** | 79.39 | 87.29 | 68.33 | 88.06 | 62.41 | 83.95 | 64.62 | 84.48 | 76.06 | 91.08 | 70.16 | 86.97 |

# I  Evaluation of Mult-turn Editing

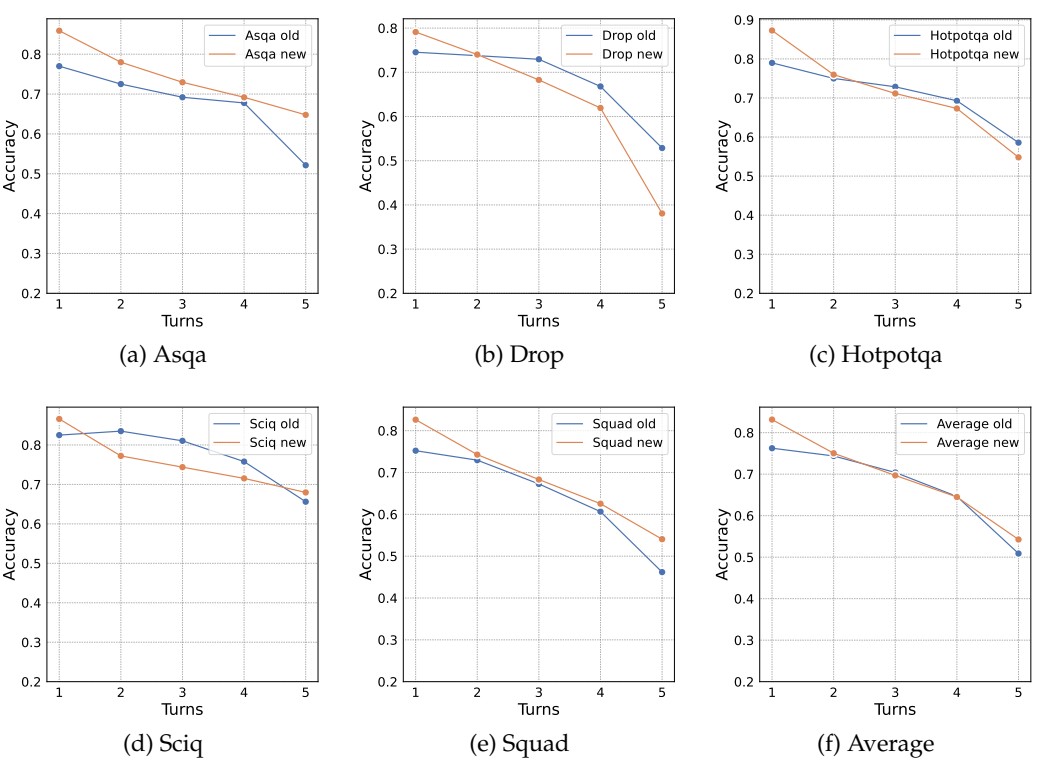

Figure 13: The evaluation result of multi-turn editing on 3B size.

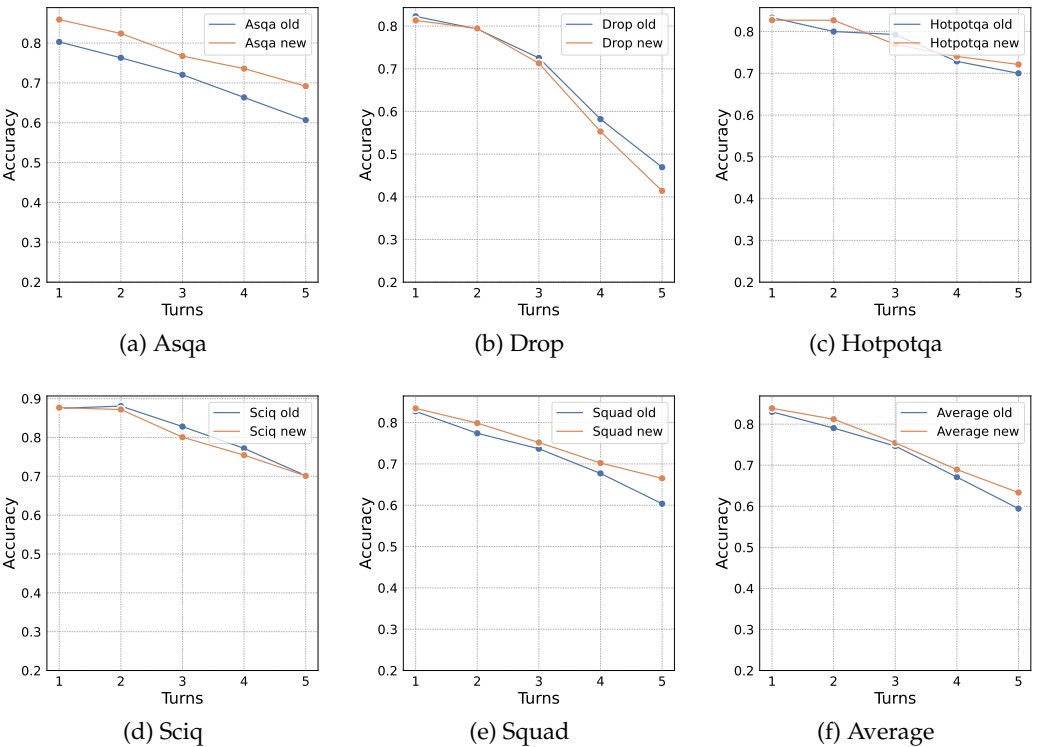

Figure 14: The evaluation result of multi-turn editing on 8B size.

## J Evaluation of Multi-chunk Ability with Edited Chunks

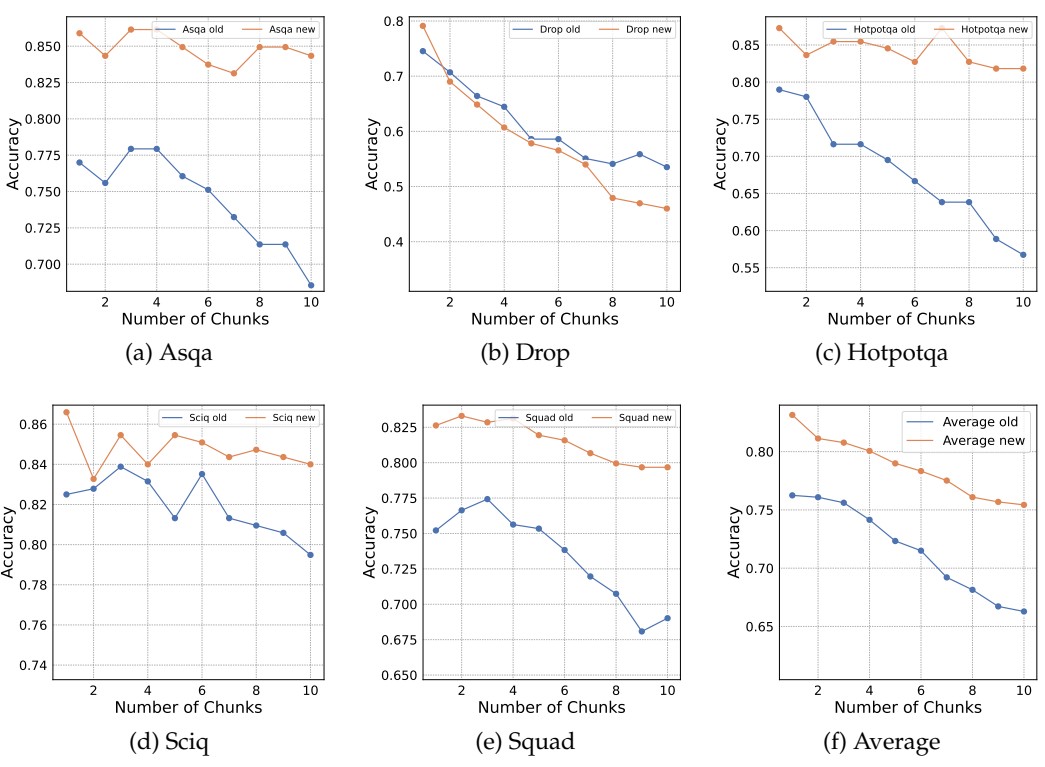

Figure 15: The evaluation results of multi-chunk on 3B size.

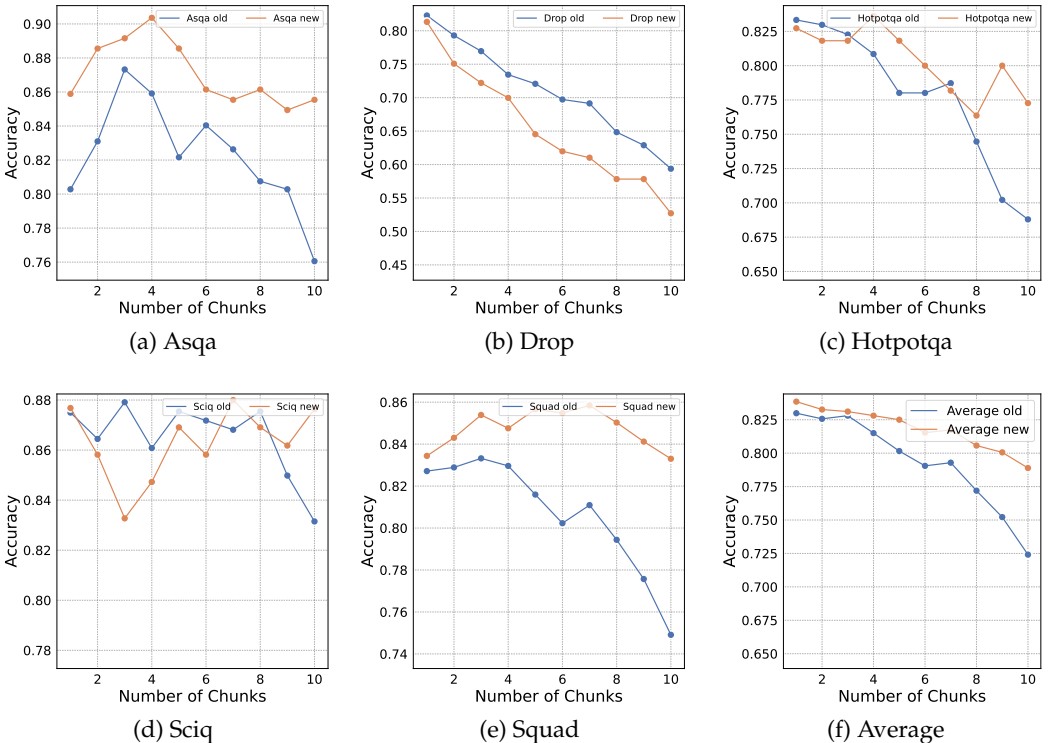

Figure 16: The evaluation result of multi-chunk on 8B size.

# K   Results on Wikipedia

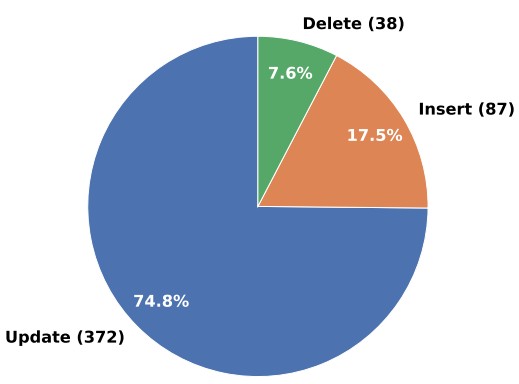

Figure 17: Wikipedia Edit Pattern Analysis.

To evaluate our method in more practical scenarios, we build a complete RAG pipeline using LlamaIndex (Liu, 2022). For the data, we construct the database of approximately 10,000 chunks from a portion of Wikipedia, where each chunk contains up to 128 tokens. To construct the editing records, we conduct a case study on Wikipedia edit histories. Specifically, we analyze the most recent 500 revisions of the "Machine Learning" entry in English Wikipedia. Each edit operation is categorized into three types: **INSERT**, **DELETE**, and **UPDATE** (Figure 17). Therefore, we construct around 1,000 editing records composed of

three types of operations, along with their corresponding questions. The proportions of the operations align with the real update distributions observed in Wikipedia.

For the model, we use BGE (Xiao et al., 2023) large as the Retriever and our 3B model to generate responses. During both the editing and querying processes, we retrieve the top-1 chunk. First, we update the database using the editing records. Specifically, we retrieve the relevant chunk using the editing content and apply the edits. After all editing records have been applied to the database, we input the questions for evaluation.

Table 12: Results of Three types of operations on WikiPedia.

| Method | UPDATE | | INSERT | | DELETE | |
|---|---|---|---|---|---|---|
| | Old (↑) | Rp (↑) | Old (↑) | New (↑) | Old (↑) | Rm (↓) |
| **Standard RAG** | | | | | | |
| LLM to edit text | 45.76 | 28.31 | **51.97** | **37.88** | **54.10** | **22.58** |
| **E$^2$-RAG w/o Editor (Efficient RAG)** | | | | | | |
| LLM to edit text | 45.23 | 30.34 | 43.31 | 32.58 | 52.46 | 29.03 |
| **E$^2$-RAG** | **45.94** | **41.02** | 41.73 | 37.12 | 52.46 | 27.42 |

