# OpenReview forum: "E$^2$-RAG: Towards Editable Efficient RAG by Editing Compressed KV Caches"
_colmweb.org/COLM/2025/Conference — COLM 2025_

### Official Review · Reviewer_ZyFk · 2025-04-28

**Rating:** 6
**Confidence:** 3
**Ethics Flag:** 1

**Summary:**

The paper introduces E²-RAG, an Editable Efficient Retrieval-Augmented Generation (RAG) method for efficiently updating compressed knowledge caches (KV caches). The proposed approach integrates an encoder-decoder architecture for efficient generation and an editor module that performs direct updates on KV caches through operations like INSERT, DELETE, and UPDATE.

**Reasons To Accept:**

* Paper is well written and easy to follow.
* The paper is written clearly, with well-structured arguments, polished figures, and well-explained datasets and tasks, making it reader-friendly.
* This research could have impact for LLM Training
* The paper provides detailed theoretical with attention map explanations the introduced methodology.

**Reasons To Reject:**

The study lacks experiments with larger-scale models (e.g., 14B or 32B), which would significantly strengthen the generalizability and practical relevance of the results.

---

> ### Author Response · Authors · 2025-05-31
> **Response to reviewer ZyFk**
>
> Thank you for your insightful feedback! We appreciate your suggestion regarding experiments with larger-scale models.
>
> We agree that evaluating E²-RAG on models in the 14B, 32B, or even larger ranges would indeed offer valuable insights into its scalability and further strengthen the generalizability and practical relevance of our findings. The decision to focus on the 3B and 8B model sizes in the current study was primarily guided by the available computational resources. We consider extending our experiments to these larger models in future work.

---

### Official Review · Reviewer_qvW3 · 2025-05-11

**Rating:** 6
**Confidence:** 3
**Ethics Flag:** 1

**Summary:**

This paper introduces E²-RAG, an editable and efficient Retrieval-Augmented Generation (RAG) framework designed to update compressed key-value (KV) caches for timely knowledge integration. Traditional RAG systems face computational bottlenecks when regenerating compressed representations after edits. E²-RAG addresses this by introducing an editor module that directly modifies KV caches using INSERT, DELETE, and UPDATE operations, inspired by database principles. The framework achieves 40x faster editing and 3x faster generation compared to baseline methods, with minimal performance degradation (1%-5%) on QA benchmarks. Experiments on diverse datasets (HotpotQA, ASQA, Drop) and models (Llama3.1-8B, Qwen2.5-7B) validate its efficiency and scalability. Ablation studies explore multi-turn editing, multi-chunk inference, and knowledge conflict resolution.

**Reasons To Accept:**

- Novel Contribution: First work to enable efficient editing of compressed KV caches in RAG, addressing a critical gap in maintaining up-to-date knowledge.

- Strong Empirical Results: Demonstrates significant speed improvements (40x editing, 3x generation) with minimal accuracy loss across multiple tasks and models.

- Comprehensive Evaluation: Includes ablation studies, multi-chunk analysis, and OOD testing, offering insights into operational robustness and limitations.

**Reasons To Reject:**

- Limited Model Generalization: Experiments are confined to Llama and Qwen models; broader validation across architectures (e.g., GPT, Mistral) is missing.

- Superficial Discussion on Conflicts: While knowledge conflicts during UPDATE/DELETE are noted, the resolution mechanisms lack depth and theoretical grounding.

---

> ### Author Response · Authors · 2025-05-31
> **Response to reviewer qvW3**
>
> Thank you for taking the time to provide insightful feedback. As stated in the Reasons To Reject, the two concerns in your comments are 1) lack of model generalization, and 2) lack of deeper theoretical grounding and discussion. Our point-wise responses are as follow:
>
> 1. **"Limited Model Generalization"**
>
> We agree that testing on a broader range of model architectures would further strengthen our findings. Our current experiments were constrained by available computational resources, but we see this as an important direction for future validation.
>
> 2. **"Superficial Discussion on Conflicts"**
>
> You are correct that while we identify and discuss knowledge conflicts arising from DELETE and UPDATE operations (e.g., in Section 6.3), the current work primarily observes their effects and hypothesizes on the editor's behavior (e.g., $\Delta V$ acting as a "mask" or "replacer"). A deeper theoretical grounding and more sophisticated resolution mechanisms are indeed significant areas that we leave for future work, as our current findings highlight the potential for further advancements in this direction.

---

> > ### Comment · Reviewer_qvW3 · 2025-06-06
> >
> > Thanks for authors' rebuttal content. After carefully reviewing the answers of weaknesses, I will keep my score.

---

### Official Review · Reviewer_rSVu · 2025-05-12

**Rating:** 5
**Confidence:** 4
**Ethics Flag:** 1

**Summary:**

The paper presents E2-RAG, a novel architecture for Retrieval-Augmented Generation (RAG) systems that emphasizes both efficiency and editability by enabling direct editing of compressed key-value (KV) caches, a challenge largely unaddressed in prior works. This is achieved via an encoder-decoder framework for compression and an editor module that supports three canonical operations—INSERT, DELETE, and UPDATE—on KV caches without needing to reprocess the entire source text.

**Questions To Authors:**

1. In line 225, the authors characterize Golden Edit as representing an upper-bound performance. However, the results reported in Table 1 indicate that the proposed method outperforms this supposed upper bound. This discrepancy raises concerns about either the definition of Golden Edit or the interpretation of the experimental results. Clarification is needed to reconcile these findings—either by refining the description of Golden Edit or by explaining under what conditions the proposed method can exceed its performance.
2. I find the distinction between the "old" and "new" settings in Table 1 somewhat unclear. Could the authors clarify what specifically differentiates these two settings, including any changes in evaluation protocol, data distribution, or editing targets?

**Reasons To Accept:**

1. This paper explores a key problem in Retrieval Augmented Generation (RAG), namely, how to update a compressed key-value cache efficiently and accurately.
2. This paper introduces a lightweight editor module equipped with a trainable LoRA adapter, enabling in-place insert, delete, and update operations on key-value caches. These editing capabilities resemble operations commonly found in relational databases and appear well-justified.
3. The authors commit to releasing code and editing datasets that could serve as valuable resources for the community to reproduce and extend this line of work.

**Reasons To Reject:**

1. Concerns about the method. While the authors argue that their method enables rapid updates to the key-value (k-v) cache and is more efficient than re-encoding the model, this claim appears counterintuitive. Specifically, the proposed editing module requires additional training, which introduces computational overhead. It is unclear how this training-based modification leads to a net efficiency gain over re-reasoning approaches that operate without such updates. Further clarification or empirical justification of this trade-off would strengthen the motivation.

2. The approach is flawed.
	- In lines 138 to 141, the authors state that the positional embedding of the 'key' should be removed during encoding. This raises the question: why is a similar operation not required for the 'value'?
	- The Editor Module appears to simply perform fine-tuning on newer data with reduced memory embedding, which seems to fall short of the ambitious motivation presented. The novelty may be somewhat overstated.
	- Would the introduction of additional key-value cache entries not negatively impact the inference speed?
3. There are concerns regarding the quality of the dataset. Although the authors used an LLM to generate a portion of the data, there appears to be no evidence of manual quality control or validation.
4. The proposed method requires model training. When comparing editing efficiency with the baseline, was the training time taken into account? (Section 5.1)
5. The methods section of this paper is not clearly written.
	- Is W_E in formula 1 the embedding parameter matrix of LLM?
	- What is $Q_apeend$ , and why can it be directly multiplied with $K$ in the key-value cache?
	- What does $O$ represent in Equation (11), and what role does it play in the proposed model?
	- The insert, delete, and update operations described in lines 162, 169, and 173 are intended solely for analyzing the role of the Editor Module and, as such, should not include method-specific components.

---

> ### Author Response · Authors · 2025-05-31
> **Response to reviewer rSVu (part1)**
>
> Thanks for your detailed reviews. Since your feedback is diverse, here please let us to address each point you have made.
>
> **For your Reasons To Reject:**
> 1. **"Concerns about the method."**
>
> We understand your concern regarding the method's efficiency and the editor module's training. But **the editor's training is an upfront investment** and the models are readily available for **direct download from open-source platforms** like Huggingface. Moreover, for scenarios with frequent updates (e.g., news, finance, or Wikipedia with 0.6 million daily edits), the speed-up of our method is substantial and quickly amortizes the initial training cost. Re-reasoning or re-encoding the entire context for every small edit is far more computationally expensive in the long run.
>
> 2. **"The approach is flawed."**
> - **"Why is a similar operation not required for the 'value'?"**
>
> This is because the **positional embedding is not applied to the 'value' in attention[1]**, there is no need to remove the positional embedding from 'value'.
>
> In Rotary Position Embedding (RoPE), **positional information is exclusively applied to the "query" and "key" vectors** within the attention mechanism.
> RoPE encodes the position information into the "query" and "key" using a rotation matrix, rather than directly adding position information to the "value". Specifically, the encoding for "query" and "key" is as follows:
>
> $q_m = R_{\Theta, m} W_q x_m$
>
> $k_n = R_{\Theta, n} W_k x_n$
>
> where $ R_{\Theta, m} $ is the rotation matrix, \$ W_q $ and $ W_k $ are the linear transformation matrices for "query" and "key", and $ x_m $ and $ x_n $ are the input hidden vectors for position $m$ and $n$.
>
> When computing the attention weights, the inner product of "query" and "key" naturally incorporates position information. Specifically, the inner product formula is:
>
> $q_m^T k_n = (R_{\Theta, m} W_q x_m)^T (R_{\Theta, n} W_k x_n)$
>
> By expanding this formula, we can see that position information is encoded into the inner product through the rotation matrices $ R_{\Theta, m} $ and $ R_{\Theta, n} $. This means that position information has already been passed to the attention weights through the interaction of "query" and "key", and there is no need to add position information to the "value".
>
> - **"The Editor Module appears to simply perform fine-tuning on newer data with reduced memory embedding"**
>
> We understand your perspective on the editor module. While it's true that our editor module performs fine-tuning on editing data with LoRA and editing embeddings, our innovation isn't centered on this. Instead, **the core novelty of E²-RAG lies in the editor's specific architectural design and its unique objective**: to directly generate offset KV caches $(\Delta K, \Delta V)$. These offset caches are then used to precisely modify existing, compressed KV caches based on our defined editing operations (INSERT, DELETE, and UPDATE). This constitutes a targeted mechanism for performing edits directly at the compressed representation level, which distinguishes it from general model fine-tuning that typically operates on raw text or aims for broader behavioral adaptations.
>
> - **"Would the introduction of additional key-value cache entries not negatively impact the inference speed?"**
>
> We appreciate the your concern. **However, the impact of adding a small number of KV cache entries from the editor module is  negligible**. Specifically, even though additional KV entries slightly increase attention computation, the number of "editing tokens" (e.g., 4 in our setup) is tiny compared to the hundreds or thousands of tokens typically processed during generation.
>
> To quantify this, we model the relative speed as inversely proportional to the total number of tokens in the KV cache:
>
> $\text{Relative Speed}= \frac{s_0}{s}$
>
> where $s_0 = 1$ is the speed when there are no KV caches. We summarize the theoretical speed impact of introducing 4 editing tokens in the table below:
>
> | KV Cache Tokens (s) | Speed (s tokens) | Speed (s+4 tokens) | Relative Speed Drop |
> | ------------------- | ---------------- | ------------------ | ------------------- |
> | 16 | 6.25% | 5% | -1.25%  |
> | 32  | 3.125%   | 2.78%   | -0.345% |
> | 64    | 1.56%    | 1.47%  | -0.09%  |
> | 128  | 0.78% | 0.758%  | -0.022%  |
> | 256     | 0.39%   | 0.385%     | -0.005%  |
> | 512     | 0.195%   | 0.194%     | -0.001%  |
>
> As shown, adding 4 tokens has a negligible impact on speed, especially when the sequence length is large (e.g., hundreds of tokens). Moreover, E²-RAG operates on compressed representations, and the primary efficiency gain comes from pre-computed KV caches, even with minor edits. This leads to up to 3× faster generation compared to standard RAG systems.
>
> ----
> [1] Roformer: Enhanced Transformer With Rotray Position Embedding

---

> ### Author Response · Authors · 2025-05-31
> **Response to reviewer rSVu (part2)**
>
> 3. **"There are concerns regarding the quality of the dataset."**
>
> We understand this point. **The quality of our datasets is ensured because they are constructed by systematically modifying established QA benchmarks**, leveraging their inherent quality for our specific editing tasks. For instance, in the INSERT operation, we reused existing questions from the original document QA, with new answers generated by Qwen2.5-7B based on the inserted knowledge. For DELETE, when necessary knowledge was removed, Qwen2.5-7B generated appropriate refusal-to-answer responses. Similarly, for UPDATE, the same language model was used to create conflicting new knowledge along with the corresponding questions and answers.
>
> 4. **"The proposed method requires model training. When comparing editing efficiency with the baseline, was the training time taken into account?"**
>
> We understand your concern. However, **the editor's training is an upfront investment. Users can directly download the models from the openscore website (Huggingface)**. So we believe the training time for the editor module is a one-off, amortized cost and is **not included in this per-edit operational efficiency metric, which is standard practice for evaluating editing/update mechanisms.**
>
> 5. **"The methods section of this paper is not clearly written."**
> - $W_E$ is indeed the token embedding matrix of the LLM, which transforms one-hot input text tokens $\mathbb{I}_X$ into dense embeddings $E$
> - $Q_{append}$ is the query of the question related to the inserted knowledge in attention, so it can perform dot product in attention.
> - $O$ represents the output of the attention when processing deleted knowledge. It describes how the attention scores $(\tilde{A}, \Delta A)$ applied to the original values $\tilde{V}$ and the offset values $\Delta V$ combine. In the DELETE operation, $\Delta V$(part of the offset KV cache from the editor) acts as a "mask" to suppress targeted segments of knowledge needed to delete.
> - These sections describe the specific objectives and mechanisms of our editor module for each defined operation (INSERT, DELETE, UPDATE). They are not merely general analyses but detail how the editor, using components like $\Delta K$, $\Delta V$, and the attention mechanism, achieves the intended edit on the KV caches. These are core to explaining how E²-RAG performs editable efficient RAG.
>
> **For your Questions To Authors:**
>
> 1. **"In line 225, the authors characterize Golden Edit as representing an upper-bound performance. However, the results reported in Table 1 indicate that ..."**
>
> We appreciate you pointing out this interesting observation. **As we have mentioned in paper, this phenomenon can be attributed to two main factors**: firstly, our E²-RAG editor's **specialized training** for optimally integrating new knowledge directly within compressed KV caches, and secondly, the inherent advantages of operating within this **simplified, compressed knowledge domain**. Specifically, the E²-RAG editor is explicitly trained to generate offset KVs $(\Delta K, \Delta V)$ that optimally integrate new information. As noted in lines 248-250 of our paper, "the KV cache for new knowledge is added after the original KV caches, bringing it closer to the query tokens, which causes the model to direct more attention toward the new knowledge". Moreover, as stated in lines 253-256, compressing context into KV caches "extracts and simplifies information, omitting less important details, which enhances the model's retrieval ability" compared to standard RAG systems that might be vulnerable to irrelevant information in a longer, albeit golden, text. This principle of the effectiveness of compressed representations is also supported by prior work[2].
>
> 2. **"I find the distinction between the "old" and "new" settings in Table 1 somewhat unclear. Could the authors clarify what specifically differentiates these two settings, including any changes in evaluation protocol, data distribution, or editing targets?"**
>
> We acknowledge the need for clarity; however, **we have provided this distinction directly in the caption of Table 1**, "We use “Old” and “New” to indicate whether the knowledge involved in the question is added through the INSERT operation or is inherent to the old chunk." **And we also have provided examples in Appendix D.**
>
> ---
>
> [2] xrag: Extreme context compression for retrieval-augmented generation with one token

---

> ### Author Response · Authors · 2025-06-05
>
> We have carefully considered your comments and detailed replies to address your concerns. We are eager to know if these responses adequately resolve the issues.
>
> Please feel free to reach out if you have additional questions. If you believe there is potential for improving the original score, we are fully willing to make any further efforts necessary.
>
> Best regards,
>
> Authors

---

> > ### Comment · Reviewer_rSVu · 2025-06-07
> >
> > Thank you for the author’s response. While part of the issue has been addressed, I still have two concerns:
> >
> > 1. In scenarios with frequent updates, your method also requires retraining. For instance, when 600,000 samples are updated, your approach must retrain all of them. Thus, the update frequency is irrelevant; what matters is comparing the cost of retraining LoRA on a single sample versus performing inference with the original model.
> >
> > 2. It remains problematic to treat editor training as a sunk upfront cost without factoring in its actual expense.
> >
> > Nonetheless, I would raise my score accordingly.

---

> > > ### Author Response · Authors · 2025-06-07
> > >
> > > Thank you once again for your constructive feedback and for raising your score. We sincerely appreciate your insightful suggestions, which have been invaluable in strengthening our work. We would like to address your remaining concern regarding the training cost of our proposed editor.
> > >
> > > We understand your primary concern is the potential training cost during deployment. Specifically, you suggested that updating a large number of samples (e.g., 600,000) would necessitate retraining our editor on this new data, which might be less efficient than simply editing the original text and re-compressing it.
> > >
> > > We would like to clarify that this retraining step is not necessary. Our editor is designed as **a general-purpose tool that does not require retraining on new data during deployment**. The key to this is its strong generalization capability, which we rigorously demonstrated in our evaluation.
> > >
> > > To ensure a fair and robust assessment, the datasets used for training and evaluation were kept completely separate:
> > >
> > > * **Training Data**: We trained the editor on samples exclusively from the training sets of HotpotQA, ASQA, and Drop.
> > > * **Evaluation Data**: We evaluated its performance on the test sets of these three datasets, as well as on two additional out-of-distribution (OOD) datasets, SciQ and SQuAD, to test its generalization limits.
> > >
> > > As presented in the paper, our results show that the editor performs exceptionally well not only on the in-distribution test sets but also demonstrates strong performance on the OOD datasets. This confirms that the editor learns a generalizable editing capability rather than memorizing patterns from the training data.
> > >
> > > Therefore, our pre-trained editor can be directly applied to the 600,000 new chunks you mentioned without any additional training. This is the foundation of our claim that the training process is a **one-time, upfront investment**. The editor is built to handle unseen data effectively, making it a practical and efficient solution for deployment.
> > >
> > > We hope this clarification fully addresses your concern. Thank you again for your valuable engagement with our paper.

---

> > ### Author Response · Authors · 2025-06-09
> >
> > As the discussion period is drawing to a close, we humbly seek your further comments on our response. We fully understand the significant demands on your time and are truly grateful for your dedication and valuable engagement with our work. We are eager to address any additional concerns you might have.
> >
> > Best regards,
> >
> > Authors

---

### Official Review · Reviewer_TEz1 · 2025-05-13

**Rating:** 6
**Confidence:** 5
**Ethics Flag:** 1

**Summary:**

This work introduces E^2 RAG that improves the efficiency of RAG by editing compressed KV caches for knowledge updates.
In this system, all the information pieces are pre-encoded as compressed KV caches and stored for retrieval.
In the inference stage, the relevant compressed KV caches will be retrieved and added into the KV cache of the LLM for generating the response. In this way, the LLM does not need to read the raw textual information on the fly and an reduced response latency could be expected.
Based on this framework, this work tries to handle the scenarios where the information that is pre-encoded compressed KV caches requires edit.
Thus, three edit operations, INSERT, DELETE, and UPDATE, to compressed KV caches are proposed in this work.
To evaluate, three datasets are constructed and three LLMs, including Llama3.2-3B, Llama3.1-8B and Qwen2.5-7B, are involved in experiments.

**Reasons To Accept:**

1. This work addresses an practical issue given the popularity of RAG.

2. Good formulation of the edit scenarios in the three edit modes.

3. The experiments include the multi-turn updates and the cases of knowledge conflicts. The results that show the cases where standard RAG performs inferior to the efficient RAG are interesting.

**Reasons To Reject:**

1. In the current framework, the top k relevant information entries will be edited for update. For a query which is relevant to n information entries and n >> k, a practical issue raises that there are (n - k) old entries that are yet to update and these old information can dominate the search results and affect the accuracy of the final RAG outcome. This work could discuss such an issue for comprehensive.
2. While the proposed method is shown effective on the two Llama models, it is less useful on the Qwen 2.5 7B model reported in Appendix H. More models could be included in experiments for comparison.

---

> ### Author Response · Authors · 2025-05-31
> **Response to reviewer TEz1**
>
> Thank you for your appreciation and detailed feedback. After carefully reading your comments, we have summarized your main concerns as: 1. Lack of analysis and discussion on impact of outdated entries related to a certain query while n >> k. 2. The need for experiments on more models, given its comparatively lower effectiveness on the Qwen 2.5 7B model. We are happy to address each of these points.
>
> 1. **The analysis and discussion on the impact of outdated entries related to a certain query when n>>k**
>
> We acknowledge the importance of investigating this scenario. To explore how models behave when facing in-contextual conflicts arising from outdated information, an experiment could be designed as follows: One might select approximately 100 questions from the test set of Squad for the three operation and duplicate the associated information chunk five times. Subsequently, the ratio of edited (updated) chunks could be varied systematically from 0.2 (1 out of 5) to 1.0 (all 5 out of 5). This would allow for an examination of model performance under varying degrees of information conflict. We provide the results for Llama-3.2 3B and Llama-3.1 8B in the table below:
>
> | Model Size | Operation      |     |      |    Ratio   |      |       |
> | :--------- | :------------- | :--------------------- | :---- | :---- | :---- | :---- |
> |            |                | 0.2                    | 0.4   | 0.6   | 0.8   | 1.0   |
> |    3B     | INSERT (↑)    | 0.4                    | 0.58  | 0.61  | 0.6   | 0.6   |
> |      3B      | DELETE (↓)  | 0.31                   | 0.34  | 0.22  | 0.19  | 0.2   |
> |       3B     | UPDATE (↑)  | 0.75                   | 0.77  | 0.82  | 0.82  | 0.82  |
> |    8B     | INSERT (↑)    | 0.69                   | 0.71  | 0.75  | 0.81  | 0.81  |
> |     8B       | DELETE (↓)  | 0.38                   | 0.39  | 0.25  | 0.16  | 0.14  |
> |      8B      | UPDATE (↑)  | 0.79                   | 0.81  | 0.82  | 0.86  | 0.86  |
>
> Firstly, all the models and operations demonstrate a similar trend where performance slightly drops when the ratio of edited chunks is reduced from 1.0 to 0.2. This is a reasonable result because when the number of edited chunks decreases, the model can be influenced by the unedited chunks, which may confuse the model and lead to incorrect answers. **However, to our surprise, our method also shows a certain level of robustness in this scenario.** Especially in the UPDATE operation, even when only 20% of the chunks are updated, the model can still notice the change and answer correctly. For instance, the performance of the 3B model remains at 0.75 when the ratio of edited chunks is only 0.2, compared to 0.82 when the ratio is 1.0.
>
> We acknowledge that updating by selecting the top-k relevant outdated entries is indeed a naive method, as it can lead to the issue of incomplete updates, as you mentioned. **However, since our work is the first to explore how to efficiently keep the RAG database up-to-date, we believe there will be future follow-up work to better address this challenge.**
>
> 2. **"It is less useful on the Qwen 2.5 7B model reported in Appendix H."**
>
> We recognize this observation. It is important to note that our computing resources are extremely limited. The Qwen 2.5 7B model, as reported, was fine-tuned for only 8,000 steps. This is a substantially lower number of training iterations compared to those applied to the Llama series models presented in our work, which likely contributes to the observed differences in performance.
>
> We provide the training loss curve in this [anonymous link](https://foxel.cc/Uploads/2025/05/4a59188b-98da-4d51-b44d-28fb6ae78412.webp).
>
> If you have any further questions, please feel free to let us know.

---

> > ### Comment · Reviewer_TEz1 · 2025-06-07
> >
> > Thank you for adding the new results. I believe the revision is more comprehensive for readers.

---

### Author Response · Authors · 2025-05-31
**General Comment**

We sincerely appreciate all reviewers' time and efforts in reviewing our paper. We are pleased to note that reviewers generally recognize our strengths:
- First work to investigate and address a practical and critical problem of maintaining up-to-date knowledge in compressed key-value caches for Efficient RAG [TEz1, rSVu, qvW3].
- Good formulation in method is reader-friendly [TEz1] and paper is clear and well-organized [ZyFk].
- Our comprehensive experimental results are interesting and insightful [TEz1, qvW3], and demonstrate that E^2-RAG delivers superior performance [qvW3].


We thank the useful suggestions from the reviewers, which help a lot in further improvement of this paper. The main revisions are summarized as follows:

**Clarifications in Methodology:**

While our paper is generally well-organized, Reviewer rSVu highlight certain points that require further clarification. Accordingly, we will revise and supplement specific details to enhance precision and understanding. We are truly thankful for these insightful suggestions, which will be instrumental in improving the paper's clarity. We welcome any further suggestions that could continue to refine the clarity of our work.

**Additional Experiments and Analysis:**
- **The impact of outdated entries related to a certain query when n>>k**

Reviewer TEz1 raised a pertinent potential issue that top-k updating will leave n-k entries remain outdated. We vary the ratio of edited chunks to investigate it.

| Model Size | Operation      |     |      |    Ratio   |      |       |
| :--------- | :------------- | :--------------------- | :---- | :---- | :---- | :---- |
|            |                | 0.2                    | 0.4   | 0.6   | 0.8   | 1.0   |
|    3B     | INSERT (↑)    | 0.4                    | 0.58  | 0.61  | 0.6   | 0.6   |
|      3B      | DELETE (↓)  | 0.31                   | 0.34  | 0.22  | 0.19  | 0.2   |
|       3B     | UPDATE (↑)  | 0.75                   | 0.77  | 0.82  | 0.82  | 0.82  |
|    8B     | INSERT (↑)    | 0.69                   | 0.71  | 0.75  | 0.81  | 0.81  |
|     8B       | DELETE (↓)  | 0.38                   | 0.39  | 0.25  | 0.16  | 0.14  |
|      8B      | UPDATE (↑)  | 0.79                   | 0.81  | 0.82  | 0.86  | 0.86  |

**The results show a certain level of robustness of our method in this scenario.** We acknowledge that updating by selecting the top-k relevant outdated entries is indeed a naive method, as it can lead to the issue of incomplete updates. **However, since our work is the first to explore how to efficiently keep the RAG database up-to-date, we believe there will be future follow-up work to better address this challenge.**

- **Experiments on a broader range of model types and larger-scale models**

Reviewers TEz1, qvW3 and ZyFk suggest that conducting experiments on a broader range of model types and larger-scale models would be valuable, a point with which we fully concur. However, due to significant limitations in available computational resources, we are unable to undertake these training for the current submission. Consequently, we have designated this valuable investigation as a direction for future work.

Please contact us if we can do something else to help you better understand and recommend our paper.

---

### Decision · Program_Chairs · 2025-07-08

**Decision:**

Accept

**Comment:**

The paper proposes E²-RAG, a novel framework that enables efficient and editable updates to compressed key-value (KV) caches in retrieval-augmented generation (RAG) systems. All four reviewers recognized the significance of the problem and the soundness of the core idea—especially the design of INSERT, DELETE, and UPDATE operations directly on KV caches. While some reviewers raised concerns about efficiency claims, training costs, and generalization to broader model families, the authors provided solid clarifications, new experiments, and in-depth responses that addressed most of these concerns. Given the paper's novelty, clear formulation, thorough experiments, and the community value of released datasets and code, I recommend acceptance.